# Robust Recovery via Implicit Bias of Discrepant Learning Rates for Double Over-parameterization

**Chong You**[†*]　　**Zhihui Zhu**[‡*]　　**Qing Qu**[♯]　　**Yi Ma**[†]

[†]Department of EECS, University of California, Berkeley
[‡]Department of Electrical and Computer Engineering, University of Denver
[♯]Center for Data Science, New York University

## Abstract

Recent advances have shown that implicit bias of gradient descent on over-parameterized models enables the recovery of low-rank matrices from linear measurements, even with no prior knowledge on the intrinsic rank. In contrast, for *robust* low-rank matrix recovery from *grossly corrupted* measurements, over-parameterization leads to overfitting without prior knowledge on both the intrinsic rank and sparsity of corruption. This paper shows that with a *double over-parameterization* for both the low-rank matrix and sparse corruption, gradient descent with *discrepant learning rates* provably recovers the underlying matrix even without prior knowledge on neither rank of the matrix nor sparsity of the corruption. We further extend our approach for the robust recovery of natural images by over-parameterizing images with deep convolutional networks. Experiments show that our method handles different test images and varying corruption levels with a single learning pipeline where the network width and termination conditions do not need to be adjusted on a case-by-case basis. Underlying the success is again the implicit bias with discrepant learning rates on different over-parameterized parameters, which may bear on broader applications. Our code is available at https://github.com/ChongYou/robust-image-recovery.

## 1  Introduction

Learning *over-parameterized models*, which have more parameters than the problem's intrinsic dimension, is becoming a crucial topic in machine learning [1–11]. While classical learning theories suggest that over-parameterized models tend to *overfit* [12], recent advances showed that an optimization algorithm may produce an *implicit bias* that regularizes the solution with desired properties. This type of results has led to new insights and better understandings on gradient descent for solving several fundamental problems, including logistic regression on linearly separated data [1], compressive sensing [2, 3], sparse phase retrieval [4], nonlinear least-squares [5], low-rank (deep) matrix factorization [6–9, 13], and deep linear neural networks [10, 11], etc.

Inspired by these recent advances [1–11], in this work we present a new type of practical methods for *robust* recovery of *structured* signals via model over-parameterization. In particular, we aim to learn an unknown signal $\mathbf{X}_\star \in \mathbb{R}^{n \times n}$ from its *grossly corrupted* linear measurements

$$\mathbf{y} = \mathcal{A}(\mathbf{X}_\star) + \mathbf{s}_\star, \tag{1}$$

---

[*]Correspondence to Chong You (cyou@berkeley.edu) and Zhihui Zhu (zhihui.zhu@du.edu). The first two authors contributed equally to this work.

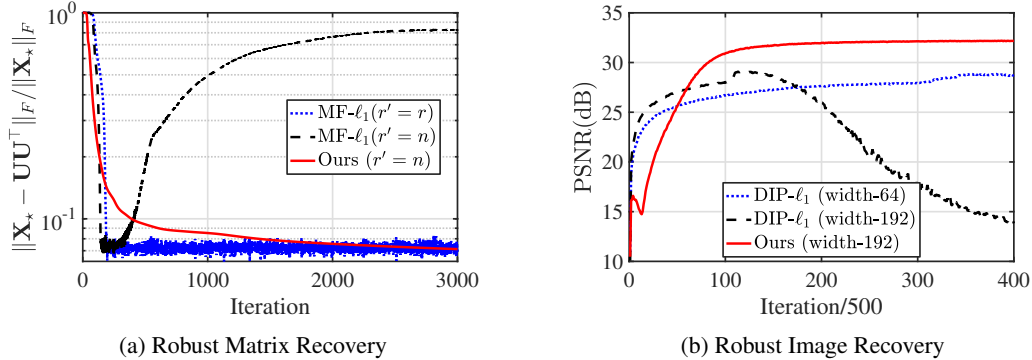

(a) Robust Matrix Recovery       (b) Robust Image Recovery

Figure 1: **Learning curves for robust recovery of low-rank matrices (a) and natural images (b).** (a) Classical matrix factorization (MF) method with $\ell_1$ penalty requires exact parameterization (left blue), otherwise over-parameterization leads to overfitting without early termination (left black). (b) Previous deep image prior (DIP) method with $\ell_1$ penalty requires tuning network width (with width = 64, right blue) or early termination (with width = 192, right black). For both problems, our double over-parameterization (DOP) method achieves superior performance and requires neither early termination nor precise parameterization (red curves).

where the linear operator $\mathcal{A}(\cdot) : \mathbb{R}^{n \times n} \mapsto \mathbb{R}^m$, and $\mathbf{s}_\star \in \mathbb{R}^m$ is a (sparse) corruption vector. Variants of the problem appear ubiquitously in signal processing and machine learning [14–19].

**Robust recovery of low-rank matrices.** The recovery of a low-rank matrix $\mathbf{X}_\star$ has broad applications in face recognition (where self-shadowing, specularity, or saturation in brightness can be modeled as outliers), video surveillance (where the foreground objects are usually modeled as outliers) and beyond [14]. A classical method for low-rank matrix recovery is via *nuclear norm* minimization[1], which is provably correct under certain incoherent conditions [14, 20]. However, minimizing nuclear norm involves expensive computations of singular value decomposition (SVD) of $n$-by-$n$ matrices [21], which prohibits its application to problem size of practical interest.

The computational challenge has been addressed by recent development of matrix factorization (MF) methods [18, 22]. Such methods are based on parameterizing the signal $\mathbf{X} \in \mathbb{R}^{n \times n}$ via the factorization $\mathbf{X} = \mathbf{U}\mathbf{U}^\top$ [23], and solving the associated nonconvex optimization problem

$$\min_{\mathbf{U} \in \mathbb{R}^{n \times r}} \left\| \mathcal{A}(\mathbf{U}\mathbf{U}^\top) - \mathbf{y} \right\|_1 . \tag{2}$$

In above, the $\ell_1$-norm penalty is adopted to account for sparse noise[2] $\mathbf{s}_\star$, and $r$ denotes the rank of $\mathbf{X}_\star$. While theoretical guarantees for the correctness of (2) can be established [18, 29–31] (see Figure 1a, dotted blue curve), the result relies on the exact information of the rank of $\mathbf{X}_\star$, which is usually *not* available *a priori* in practice. On the other hand, if $\mathbf{U} \in \mathbb{R}^{n \times r}$ in (2) is replaced with an *over-parameterized* factor $\mathbf{U} \in \mathbb{R}^{n \times n}$, for which $r$ is no longer required, the method easily *overfits* the corruptions and does not produce the desired solution $\mathbf{X}_\star$, unless we do some *early stopping* [32, 33] (see Figure 1a, dashed black curve).

**Robust recovery of natural images.** Robust recovery of natural images $\mathbf{X}_\star$ is often considered as a challenging task due to the lack of *universal* mathematical modeling for natural images. While sparse and low-rank based methods have been demonstrated to be effective for years [34–39], the state-of-the-art performance is obtained by learned priors with deep convolutional neural networks. Such methods operate by end-to-end training of neural networks from pairs of corrupted/clean images [40–42], and often *cannot* effectively handle test cases with corruption type and noise level that are different from those of the training data.

Recently, this challenge has been partially addressed by the so-called *deep image prior* (DIP) [43], which has shown impressive results on many image reconstruction tasks. The method is based on parameterizing an image $\mathbf{X}$ by a deep network $\mathbf{X} = \phi(\boldsymbol{\theta})$, which turns out to be a flexible prior for

modeling the underlying distributions of a variety of natural images. The network $\phi(\boldsymbol{\theta})$ has a U-shaped architecture and may be viewed as a multi-layer, nonlinear extension of the low-rank matrix factorization $\mathbf{U}\mathbf{U}^\top$. Hence, it may not come as a surprise that DIP inherits the drawbacks of the MF approaches for low-rank matrix recovery. Namely, it requires either a meticulous choice of network width [44] (see Figure 1b, dotted blue curve) or early termination of the learning process [33] (see Figure 1b, dashed black curve).

**Overview of our methods and contributions.** We introduce a *double over-arameterization* (**DOP**) framework for robust recovery that effectively addresses the issues associated with MF and DIP. For the low-rank matrix recovery problem, our method is based on over-parameterizing both the low-rank matrix $\mathbf{X}_\star$ (with matrix factorization) and the sparse corruption $\mathbf{s}_\star$ (with Hadamard product). This gives rise to a highly over-parameterized formulation that has infinitely many global solutions. We further present a gradient descent algorithm with *discrepant learning rates* on different optimization variables, and show that it introduces an implicit algorithmic bias that enables correct recovery of $(\mathbf{X}_\star, \mathbf{s}_\star)$ with provable guarantees. The benefit of such an approach *w.r.t.* the state of the art is summarized as follows (see also Table 1):

- *More scalable.* Our method is based on gradient descent only and does not require computing an SVD per iteration as in convex approaches [14]. Hence it is significantly more scalable.
- *Prior knowledge free.* Our method requires *no* prior knowledge on neither the rank of $\mathbf{X}_\star$ nor the sparsity of $\mathbf{s}_\star$, and is free of parameter tuning. This is *in contrast* to existing nonconvex approaches [16, 19, 30, 45] for which the true rank and/or sparsity are required *a priori*.
- *Provably correct.* Under similar conditions of the convex approach, our method converges to the ground truth $(\mathbf{X}_\star, \mathbf{s}_\star)$ asymptotically.

Underlying the success of our method is the notion of *implicit bias of discrepant learning rates*. The idea is that the algorithmic low-rank and sparse regularizations need to be balanced for the purpose of identifying the underlying rank and sparsity. With a lack of means of tuning a regularization parameter, we show in Section 2.3 that

Table 1: Comparison of different approaches to matrix recovery.

| Methods | Convex (Eq. (8)) | Nonconvex (Eq. (2)) | Ours (Eq. (3)) |
|---|---|---|---|
| Provably correct? | ✓ | ✓ | ✓ |
| Prior knowledge free? | ✓ | | ✓ |
| Scalable? | | ✓ | ✓ |

the desired balance can be obtained by using *different* learning rates for different optimization parameters. Such a property may be of separate interest and bear on a broader range of problems.

Finally, we demonstrate the broad applicability of the DOP framework for the task of the robust recovery of natural images. We only need to replace the over-parameterization $\mathbf{U}\mathbf{U}^\top$ for low-rank matrices by a U-shaped network $\phi(\boldsymbol{\theta})$ for natural images (as adopted in DIP [43]) and solve the optimization problem by gradient descent with discrepant learning rates. This produces a powerful and easy-to-use method with favorable properties when compared to the original DIP (see Figure 1b). To be precise, our method handles different test images and varying corruption levels with a single learning pipeline, in which network width and termination conditions do not need to be adjusted on a case-by-case basis.

## 2 Main Results and Algorithms

### 2.1 A Double Over-Parameterization (DOP) Formulation

As precluded in (1), we first consider the problem of recovering a rank-$r$ $(r \ll n)$ positive semidefinite matrix[3] $\mathbf{X}_\star \in \mathbb{R}^{n \times n}$ from its corrupted linear measurements $\mathbf{y} = \mathcal{A}(\mathbf{X}_\star) + \mathbf{s}_\star$, where $\mathbf{s}_\star \in \mathbb{R}^m$ is a vector of sparse corruptions. Recent advances on algorithmic implicit bias for optimizing over-parameterized models [2, 3, 6–9] motivate us to consider a nonlinear least squares for robust matrix recovery, with over-parameterization $\mathbf{X} = \mathbf{U}\mathbf{U}^\top$ and $\mathbf{s} = \mathbf{g} \circ \mathbf{g} - \mathbf{h} \circ \mathbf{h}$:

$$\min_{\mathbf{U}\in\mathbb{R}^{n\times r'},\{\mathbf{g},\mathbf{h}\}\subseteq\mathbb{R}^m} f(\mathbf{U}, \mathbf{g}, \mathbf{h}) := \frac{1}{4}\left\|\mathcal{A}\left(\mathbf{U}\mathbf{U}^\top\right) + (\mathbf{g} \circ \mathbf{g} - \mathbf{h} \circ \mathbf{h}) - \mathbf{y}\right\|_2^2, \quad (3)$$

where the dimensional parameter $r' \geq r$ and "$\circ$" denotes the Hadamard (i.e., entrywise) product. In practice, the choice of $r'$ depends on how much prior information we have for $\mathbf{X}_\star$: $r'$ can be either taken as an estimated upper bound for $r$, or taken as $r' = n$ with no prior knowledge. In the following, we provide more backgrounds for the choice of our formulation (3).

- *Implicit low-rank prior via matrix factorization.* For the *vanilla* low rank matrix recovery problem with no outlier (i.e., $\mathbf{s}_\star = \mathbf{0}$), the low-rank matrix $\mathbf{X}_\star$ can be recovered via over-parameterization $\mathbf{X} = \mathbf{U}\mathbf{U}^\top$ [6–9, 48]. In particular, the work [6] showed that with small initialization and infinitesimal learning rate, gradient descent on

$$\min_{\mathbf{U} \in \mathbb{R}^{n \times n}} \frac{1}{2} \left\| \mathcal{A}(\mathbf{U}\mathbf{U}^\top) - \mathbf{y} \right\|_2^2 \tag{4}$$

converges to the minimum nuclear norm solution under certain commute conditions for $\mathcal{A}(\cdot)$.

- *Implicit sparse prior via Hadamard multiplication.* For the classical sparse recovery problem [49, 50] which aims to recover a sparse $\mathbf{s}_\star \in \mathbb{R}^m$ from its linear measurement $\mathbf{b} = \mathbf{A}\mathbf{s}_\star$, recent work [2, 3] showed that it can also be dealt with via over-parameterization $\mathbf{s} = \mathbf{g} \circ \mathbf{g} - \mathbf{h} \circ \mathbf{h}$. In particular, the work [2] showed that with small initialization and infinitesimal learning rate, gradient descent on

$$\min_{\{\mathbf{g},\mathbf{h}\} \subseteq \mathbb{R}^m} \left\| \mathbf{A}(\mathbf{g} \circ \mathbf{g} - \mathbf{h} \circ \mathbf{h}) - \mathbf{b} \right\|_2^2 \tag{5}$$

correctly recovers the sparse $\mathbf{s}_\star$ when $\mathbf{A}$ satisfies certain restricted isometry properties [51].

The benefit of DOP in (3), as we shall see, is that it allows specific algorithms to *automatically* identify both the intrinsic rank of $\mathbf{X}_\star$ and the sparsity level of $\mathbf{s}_\star$ without any prior knowledge.

## 2.2 Algorithmic Regularizations via Gradient Descent

Obviously, over-parameterization leads to *under-determined* problems which can have infinite number of solutions, so that not all solutions of (3) correspond to the desired $(\mathbf{X}_\star, \mathbf{s}_\star)$. For example, for any given $\mathbf{U}$, one can always construct a pair $(\mathbf{g}, \mathbf{h})$ for (3) such that $(\mathbf{U}, \mathbf{g}, \mathbf{h})$ achieves the global minimum value $f = 0$. Nonetheless, as we see in this work, the gradient descent iteration on (3),

$$\mathbf{U}_{k+1} = \mathbf{U}_k - \tau \cdot \nabla_{\mathbf{U}} f(\mathbf{U}_k, \mathbf{g}_k, \mathbf{h}_k) = \mathbf{U}_k - \tau \cdot \mathcal{A}^*(\mathbf{r}_k)\mathbf{U}_k,$$

$$\begin{bmatrix} \mathbf{g}_{k+1} \\ \mathbf{h}_{k+1} \end{bmatrix} = \begin{bmatrix} \mathbf{g}_k \\ \mathbf{h}_k \end{bmatrix} - \alpha \cdot \tau \cdot \begin{bmatrix} \nabla_{\mathbf{g}} f(\mathbf{U}_k, \mathbf{g}_k, \mathbf{h}_k) \\ \nabla_{\mathbf{h}} f(\mathbf{U}_k, \mathbf{g}_k, \mathbf{h}_k) \end{bmatrix} = \begin{bmatrix} \mathbf{g}_k \\ \mathbf{h}_k \end{bmatrix} - \alpha \cdot \tau \cdot \begin{bmatrix} \mathbf{r}_k \circ \mathbf{g}_k \\ -\mathbf{r}_k \circ \mathbf{h}_k \end{bmatrix}, \tag{6}$$

with properly chosen learning rates $(\tau, \ \alpha \cdot \tau)$ enforces implicit bias on the solution path, that it automatically identifies the desired, regularized solution $(\mathbf{X}_\star, \mathbf{s}_\star)$. Here, in (6) we have $\mathcal{A}^*(\cdot)$ : $\mathbb{R}^m \mapsto \mathbb{R}^{n \times n}$ being the conjugate operator of $\mathcal{A}(\cdot)$ and $\mathbf{r}_k = \mathcal{A}(\mathbf{U}_k\mathbf{U}_k^\top) + \mathbf{g}_k \circ \mathbf{g}_k - \mathbf{h}_k \circ \mathbf{h}_k - \mathbf{y}$.

It should be noted that the scalar $\alpha > 0$, which controls the ratio of the learning rates for $\mathbf{U}$ and $(\mathbf{g}, \mathbf{h})$, plays a crucial role on the quality of the solution that the iterate in (6) converges to (see Figure 2a). We will discuss this in more details in the next subsection (i.e., Section 2.3).

**Convergence to low-rank & sparse solutions.** Based on our discussion in Section 2.1, it is expected that the gradient descent (6) converges to a solution $(\mathbf{U}, \mathbf{g}, \mathbf{h})$ such that

$$\mathbf{X} = \mathbf{U}\mathbf{U}^\top \qquad \text{and} \qquad \mathbf{s} = \mathbf{g} \circ \mathbf{g} - \mathbf{h} \circ \mathbf{h} \tag{7}$$

have the minimum nuclear norm and $\ell_1$ norm, respectively. More specifically, we expect the solution $(\mathbf{X}, \mathbf{s})$ in (7) of the nonlinear least squares (3) also serves as a *global* solution to a convex problem

$$\min_{\mathbf{X} \in \mathbb{R}^{n \times n}, \ \mathbf{s} \in \mathbb{R}^m} \|\mathbf{X}\|_* + \lambda \cdot \|\mathbf{s}\|_1, \quad \text{s.t.} \quad \mathcal{A}(\mathbf{X}) + \mathbf{s} = \mathbf{y}, \ \mathbf{X} \succeq \mathbf{0}, \tag{8}$$

for which we state more rigorously in Section 3 under a particular setting. However, it should be noted that obtaining the global solution of (8) does not necessarily mean that we find the desired solution $(\mathbf{X}_\star, \mathbf{s}_\star)$ — the penalty $\lambda > 0$ in (8), which balances the low-rank and the sparse regularizations, plays a crucial role on the quality of the solution to (8). For instance, when $\mathcal{A}(\cdot)$ is the identity operator, under proper conditions of $(\mathbf{X}_\star, \mathbf{s}_\star)$, the work [14] showed that the global solution of (8) is exactly the target solution $(\mathbf{X}_\star, \mathbf{s}_\star)$ *only* when $\lambda = 1/\sqrt{n}$.

## 2.3 Implicit Bias with Discrepant Learning Rates

As noted above, a remaining challenge is to control the implicit regularization of the gradient descent (6) so that the algorithm converges to the solution to (8) with the desired value $\lambda = 1/\sqrt{n}$. Without explicit regularization terms in our new objective (3), at first glance this might seem impossible. Nonetheless, we show that this can simply be achieved by adapting the ratio of learning rates $\alpha$ between $\mathbf{U}$ and $(\mathbf{g}, \mathbf{h})$ in our gradient step (6), which is one of our key contributions in this work that could also bear broader interest. More specifically, this phenomenon can be captured by the following observation.

**Observation.** *With small enough learning rate $\tau$ and initialization $(\mathbf{U}_0, \mathbf{g}_0, \mathbf{h}_0)$ close enough to the origin, gradient descent (6) converges to a solution of (8) with $\lambda = 1/\alpha$.*

In comparison to the classical optimization theory [52] where learning rates *only* affect algorithm convergence rate but not the quality of the solution, our observation here is surprisingly different — using discrepant learning rates on different over-parameterized variables actually affects the specific solution that the algorithm converges to[4]. In the following, let us provide some intuitions of why this happens and discuss its implications. We leave a more rigorous mathematical treatment to Section 3.

**Intuitions.** The relation $\lambda = 1/\alpha$ implies that a large learning rate for one particular optimization variable in (6) leads to a small penalty on its implicit regularization counterpart in (8). From a high-level perspective, this happens because a larger learning rate allows the optimization variable to move faster away from its initial point (which is close to the origin), resulting in a weaker regularization effect (which penalizes the distance of the variable to the origin) on its solution path.

**Implications.** The implicit bias of discrepant learning rates provides a new and powerful way for controlling the regularization effects in over-parametrized models. For robust matrix recovery in particular, it reveals an equivalence between our method in (3) and the convex method in (8) with one-to-one correspondence between learning rate ratio $\alpha$ and the regularization parameter $\lambda$. By picking $\alpha = \sqrt{n}$, we may directly quote results from [14] and conclude that our method correctly recovers $\mathbf{X}_\star$ with information-theoretically optimal sampling complexity and sparsity levels (see Figure 2). Note that this is achieved with no prior knowledge on the rank of $\mathbf{X}_\star$ and sparsity of $\mathbf{s}_\star$. Next, we show that such benefits have implications beyond robust low-rank matrix recovery.

## 2.4 Extension to Robust Recovery of Natural Images

Finally, we address the problem of robust recovery of a natural image[5] $\mathbf{X}_\star \in \mathbb{R}^{C \times H \times W}$ from its corrupted observation $\mathbf{y} = \mathbf{X}_\star + \mathbf{s}_\star$, for which the structure of $\mathbf{X}_\star$ cannot be fully captured by a low-rank model. Inspired by the work [43] on showing the effectiveness of an image prior from a deep convolutional network $\mathbf{X} = \phi(\boldsymbol{\theta})$ of particular architectures, where $\boldsymbol{\theta} \in \mathbb{R}^c$ denotes network parameters, we use the following formulation for image recovery:

$$\min_{\boldsymbol{\theta} \in \mathbb{R}^c, \, \{\mathbf{g}, \mathbf{h}\} \subseteq \mathbb{R}^{C \times H \times W}} f(\boldsymbol{\theta}, \mathbf{g}, \mathbf{h}) = \frac{1}{4} \|\phi(\boldsymbol{\theta}) + (\mathbf{g} \circ \mathbf{g} - \mathbf{h} \circ \mathbf{h}) - \mathbf{y}\|_F^2. \tag{9}$$

As we will empirically demonstrate in Section 4.2, gradient descent on (9)

$$\boldsymbol{\theta}_{k+1} = \boldsymbol{\theta}_k - \tau \cdot \nabla_{\boldsymbol{\theta}} f(\boldsymbol{\theta}_k, \mathbf{g}_k, \mathbf{h}_k),$$
$$\begin{bmatrix} \mathbf{g}_{k+1} \\ \mathbf{h}_{k+1} \end{bmatrix} = \begin{bmatrix} \mathbf{g}_k \\ \mathbf{h}_k \end{bmatrix} - \alpha \cdot \tau \cdot \begin{bmatrix} \nabla_{\mathbf{g}} f(\boldsymbol{\theta}_k, \mathbf{g}_k, \mathbf{h}_k) \\ \nabla_{\mathbf{h}} f(\boldsymbol{\theta}_k, \mathbf{g}_k, \mathbf{h}_k) \end{bmatrix}, \tag{10}$$

with a balanced learning rate ratio $\alpha$ also enforces implicit regularizations on the solution path to the desired solution. It should be noted that this happens even that the over-parameterization $\mathbf{X} = \phi(\boldsymbol{\theta})$ is a highly *nonlinear* network (in comparison with shallow linear network $\mathbf{X} = \mathbf{U}\mathbf{U}^\top$ [6] or deep linear network [8]), which raises several intriguing theoretical questions to be further investigated. For example, we empirically observe that the effect of $\alpha$ is analogous to that in low-rank matrix recovery, meaning that the optimal $\alpha$ is independent of the image $\mathbf{X}_\star$ and corruption level of $\mathbf{s}_\star$. This provides an attractive feature of our method as it does not require any parameter tuning.

# 3 Convergence Analysis of Gradient Flow Dynamics

We provide a dynamical analysis certifying our observation in Section 2.3. Similar to [6, 8], we consider a special case where the measurement matrices $\{\mathbf{A}_i\}_{i=1}^m$ associated with $\mathcal{A}$ commute[6], and investigate the trajectories of the discrete gradient iterate of $\mathbf{U}_k$, $\mathbf{g}_k$, and $\mathbf{h}_k$ in (6) with *infinitesimal* learning rate $\tau$ (i.e., $\tau \to 0$). In other words, we study their *continuous* dynamics counterparts $\mathbf{U}_t(\gamma)$, $\mathbf{g}_t(\gamma)$, and $\mathbf{h}_t(\gamma)$, which are parameterized by $t \in [0, +\infty)$ and initialized at $t = 0$ with

$$\mathbf{U}_0(\gamma) = \gamma \mathbf{I}, \quad \mathbf{g}_0(\gamma) = \gamma \mathbf{1}, \quad \mathbf{h}_0(\gamma) = \gamma \mathbf{1}, \tag{11}$$

where $\gamma > 0$. Thus, analogous to (6), the behaviors of the continuous gradient flows can be captured via the following differential equations

$$\dot{\mathbf{U}}_t(\gamma) = \lim_{\tau \to 0} (\mathbf{U}_{t+\tau}(\gamma) - \mathbf{U}_t(\gamma))/\tau = -\mathcal{A}^*(\mathbf{r}_t(\gamma))\mathbf{U}_t(\gamma), \tag{12}$$

$$\begin{bmatrix} \dot{\mathbf{g}}_t(\gamma) \\ \dot{\mathbf{h}}_t(\gamma) \end{bmatrix} = \lim_{\tau \to 0} \left( \begin{bmatrix} \mathbf{g}_{t+\tau}(\gamma) \\ \mathbf{h}_{t+\tau}(\gamma) \end{bmatrix} - \begin{bmatrix} \mathbf{g}_t(\gamma) \\ \mathbf{h}_t(\gamma) \end{bmatrix} \right) / \tau = -\alpha \cdot \begin{bmatrix} \mathbf{r}_t(\gamma) \circ \mathbf{g}_t(\gamma) \\ -\mathbf{r}_t(\gamma) \circ \mathbf{h}_t(\gamma), \end{bmatrix}, \tag{13}$$

with $\mathbf{r}_t(\gamma) = \mathcal{A}(\mathbf{U}_t(\gamma)\mathbf{U}_t^\top(\gamma)) + \mathbf{g}_t(\gamma) \circ \mathbf{g}_t(\gamma) - \mathbf{h}_t(\gamma) \circ \mathbf{h}_t(\gamma) - \mathbf{y}$. Let $\mathbf{X}_t(\gamma) = \mathbf{U}_t(\gamma)\mathbf{U}_t^\top(\gamma)$, then we can derive the gradient flow for $\mathbf{X}_t(\gamma)$ via chain rule

$$\dot{\mathbf{X}}_t(\gamma) = \dot{\mathbf{U}}_t(\gamma)\mathbf{U}_t^\top(\gamma) + \mathbf{U}_t(\gamma)\dot{\mathbf{U}}_t^\top(\gamma) = -\mathcal{A}^*(\mathbf{r}_t(\gamma))\mathbf{X}_t(\gamma) - \mathbf{X}_t(\gamma)\mathcal{A}^*(\mathbf{r}_t(\gamma)). \tag{14}$$

For any $\gamma > 0$, suppose the limits of $\mathbf{X}_t(\gamma)$, $\mathbf{g}_t(\gamma)$, and $\mathbf{h}_t(\gamma)$ as $t \to +\infty$ exist and denote by

$$\mathbf{X}_\infty(\gamma) := \lim_{t \to +\infty} \mathbf{X}_t(\gamma), \quad \mathbf{g}_\infty(\gamma) := \lim_{t \to +\infty} \mathbf{g}_t(\gamma), \quad \mathbf{h}_\infty(\gamma) := \lim_{t \to +\infty} \mathbf{h}_t(\gamma). \tag{15}$$

Then when the initialization is infinitesimally small with $\gamma \to 0$, we show that the following holds.

**Theorem 1.** *Assume that the measurement matrices $\{\mathbf{A}_i\}_{i=1}^m$ are symmetric and commutable with $\mathbf{A}_i\mathbf{A}_j = \mathbf{A}_j\mathbf{A}_i$ for $\forall\, 1 \le i \ne j \le m$, and the gradient flows of $\mathbf{U}_t(\gamma)$, $\mathbf{g}_t(\gamma)$, and $\mathbf{h}_t(\gamma)$ satisfy (12) and (13) and are initialized by (11). Let $\mathbf{X}_\infty(\gamma)$, $\mathbf{g}_\infty(\gamma)$, and $\mathbf{h}_\infty(\gamma)$ be the limit points as defined in (15). Suppose that our initialization is infinitesimally small such that*

$$\widehat{\mathbf{X}} := \lim_{\gamma \to 0} \mathbf{X}_\infty(\gamma), \quad \widehat{\mathbf{g}} := \lim_{\gamma \to 0} \mathbf{g}_\infty(\gamma), \quad \widehat{\mathbf{h}} := \lim_{\gamma \to 0} \mathbf{h}_\infty(\gamma)$$

*exist and $(\widehat{\mathbf{X}}, \widehat{\mathbf{g}}, \widehat{\mathbf{h}})$ is a global optimal solution to (3) with*

$$\mathcal{A}(\widehat{\mathbf{X}}) + \widehat{\mathbf{s}} = \mathbf{y} \quad and \quad \widehat{\mathbf{s}} = \widehat{\mathbf{g}} \circ \widehat{\mathbf{g}} - \widehat{\mathbf{h}} \circ \widehat{\mathbf{h}}.$$

*Then we have $\widehat{\mathbf{g}} \circ \widehat{\mathbf{h}} = \mathbf{0}$, and $(\widehat{\mathbf{X}}, \widehat{\mathbf{s}})$ is also a global optimal solution to (8), with $\lambda = \alpha^{-1}$.*

A proof of Theorem 1 is given in Appendix B. Our proof follows a similar procedure as that in [6] and is based on constructing a dual certificate for the convex problem in (8) with the algorithm dynamic of gradient descent. The major difference from [6] is on handling the additional sparse term $\mathbf{s}_\star$ and characterizing the discrepant learning rates in the verification of the dual certificate.

Our result shows that among the infinite many global solutions to (3), gradient descent biases towards the one with the minimum nuclear norm (for $\mathbf{X}$) and $\ell_1$ norm (for $\mathbf{s}$) with relative weight controlled by $\alpha$. In the following, we discuss the rationality of the assumptions for Theorem 1.

- Since gradient descent almost surely converges to a local minimum [56] and any local minimum is likely to be a global minimum in low-rank matrix optimization with over-parameterization (i.e., $\mathbf{U} \in \mathbb{R}^{n \times n}$) [57], the assumption that $(\widehat{\mathbf{X}}, \widehat{\mathbf{g}}, \widehat{\mathbf{h}})$ is a global optimal solution is generally satisfied.

- The commutative assumption on $\mathcal{A}$ is commonly adopted in the analysis of over-parameterized low-rank matrix recovery. While it may rule out many practical cases (e.g., when $\mathcal{A}$ is identity), it is empirically observed to be non-essential [6, 8]. A recent work [7] provides a more sophisticated analysis of the discrete dynamic under the restricted isometric assumption where the commutative assumption is not needed. We believe such analysis can be extended to our DOP setting as well, and leave it as future work.

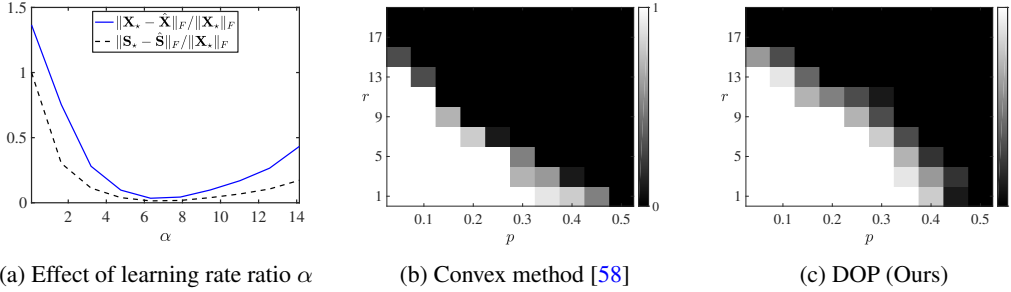

(a) Effect of learning rate ratio $\alpha$     (b) Convex method [58]     (c) DOP (Ours)

Figure 2: **Numerical results for robust PCA** (with $n = 50$). (a) Relative reconstruction error for the output $(\widehat{\mathbf{X}}, \widehat{\mathbf{S}})$ of our method with varying step size ratio $\alpha$. Here, we set $r = 5$ and $p = 30\%$. (b, c) Probability of successful recovery over 10 trials with varying $r$ (y-axis) and $p$ (x-axis). Here, we fixed $\alpha = \sqrt{n}$. White indicates always successful recovery, while black means total failure.

## 4 Experiments

In this section, we provide experimental evidence for the implicit bias of the learning rate discussed in Section 2.3. Through experiments for low-rank matrix recovery, Section 4.1 shows that the learning rate ratio $\alpha$ affects the solution that gradient descent converges to, and that an optimal $\alpha$ does not depend on the rank of matrix and sparsity of corruption. Furthermore, Section 4.2 shows the effectiveness of implicit bias of learning rate in alleviating overfitting for robust image recovery, and demonstrates that our method produces better recovery quality when compared to DIP for varying test images and corruption levels, all with a single model and set of learning parameters.

### 4.1 Robust Recovery of Low-rank Matrices

We conduct experiments for a particular case of low-rank matrix recovery problem, namely the robust PCA problem, in which the operator $\mathcal{A}$ is the identity operator. More specifically, the goal is to recovery a low-rank matrix $\mathbf{X}_\star$ and a sparse matrix $\mathbf{S}_\star$ from the mixture $\mathbf{Y} = \mathbf{X}_\star + \mathbf{S}_\star$, possibly without prior knowledge on the rank $r$ of $\mathbf{X}_\star$ and the percentage of nonzero entries $p$ of $\mathbf{S}_\star$. For any given $r$, we generate $\mathbf{X}_\star \in \mathbb{R}^{n \times n}$ by setting $\mathbf{X}_\star = \mathbf{U}_\star \mathbf{U}_\star^\top$, where $\mathbf{U}_\star$ is an $n \times r$ matrix with i.i.d. entries drawn from standard Gaussian distribution. For any given $p$, we generate $\mathbf{S}_\star \in \mathbb{R}^{n \times n}$ by sampling uniformly at random $n^2 \cdot p$ locations from the matrix and setting those entries by sampling i.i.d. from a zero-mean Gaussian distribution with standard deviation 10. We use $n = 50$.

We apply our DOP method in (3) for the robust PCA problem, Specifically, we initialize $\mathbf{U}$ and $\mathbf{g}$ by drawing i.i.d. entries from zero mean Gaussian distribution with standard deviation $10^{-4}$, and initialize $\mathbf{h}$ to be the same as $\mathbf{g}$. The learning rate for $\mathbf{U}$ as well as for $\{\mathbf{g}, \mathbf{h}\}$ are set to $\tau$ and $\alpha \cdot \tau$, respectively, where $\tau = 10^{-4}$ for all experiments.

**Effect of learning rate ratio $\alpha$.** We set $r = 5$ and $p = 30\%$, perform $2 \times 10^4$ gradient descent steps and compute reconstruction errors for the output of our algorithm $(\widehat{\mathbf{X}}, \widehat{\mathbf{S}})$ relative to the ground truth $(\mathbf{X}_\star, \mathbf{S}_\star)$. Figure 2a illustrates the performance with varying values of $\alpha$. We observe that $\alpha$ affects the solution that the algorithm converges to, which verifies that the learning rate ratio has an implicit regularization effect. Moreover, the best performance is given by $\alpha = \sqrt{n}$, which is in accordance with our theoretical result in Theorem 1.

**Effect of rank $r$ and outlier ratio $p$ and phase transition.** We now fix $\alpha = \sqrt{n}$ and study the ability of our method for recovering matrices of varying rank $r$ with varying percentage of corruption $p$. For each pair $(r, p)$, we apply our method and declare the recovery to be successful if the relative reconstruction error $\frac{\|\widehat{\mathbf{X}} - \mathbf{X}_\star\|_F}{\|\mathbf{X}_\star\|_F}$ is less than 0.1. Figure 2c displays the fraction of successful recovery in 10 Monte Carlo trials. It shows that a single value of the parameter $\alpha$ leads to correct recovery for a wide range of $r$ and $p$. Figure 2b shows the fraction of successful recovery for the convex method in (8) (with the Accelerated Proximal Gradient [58] solver[7]).

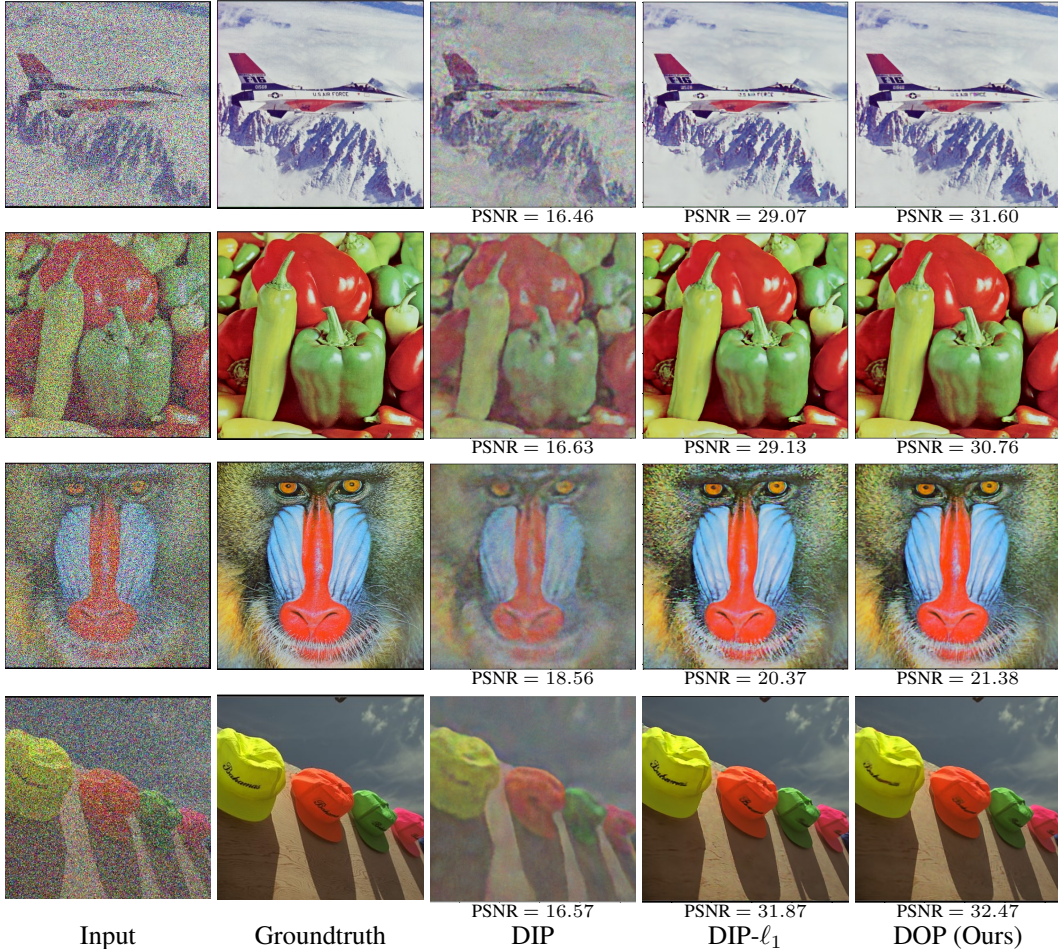

|  |  | PSNR = 16.46 | PSNR = 29.07 | PSNR = 31.60 |
|  |  | PSNR = 16.63 | PSNR = 29.13 | PSNR = 30.76 |
|  |  | PSNR = 18.56 | PSNR = 20.37 | PSNR = 21.38 |
|  |  | PSNR = 16.57 | PSNR = 31.87 | PSNR = 32.47 |
| Input | Groundtruth | DIP | DIP-$\ell_1$ | DOP (Ours) |

Figure 3: **Robust image recovery for salt-and-pepper corruption.** PSNR of the results is displayed below the images. For our method, all cases use the same network width, learning rate, and termination condition. For DIP and DIP-$\ell_1$, case-dependent early stopping is used which is essential for their good performance. Despite that, DOP (our method) achieves the highest PSNRs and best visual quality.

## 4.2 Robust Recovery of Natural Images

Following [43], we evaluate the performance of our method for robust image recovery using four images, namely "F16", "Peppers", "Baboon" and "Kodim03" (see the second column of Figure 3), from a standard dataset[8]. Corruption for the images is synthesized by adding salt-and-pepper noise, where ratio $p$ of randomly chosen pixels are replaced with either 1 or 0 (each with $50\%$ probability).

The network $\phi(\boldsymbol{\theta})$ for our method in (9) is the same as the denoising network in [43], which has a U-shaped architecture with skip connections. Each layer of the network contains a convolutional layer, a nonlinear LeakyReLU layer and a batch normalization layer. We also follow [43] on the initialization of network parameters $\boldsymbol{\theta}$, which is the Kaiming initialization. Meanwhile, we initialize $\mathbf{g}$ and $\mathbf{h}$ by drawing i.i.d. entries from zero-mean Gaussian distribution with a standard deviation of $10^{-5}$. The learning rate $\tau$ is set to $1.0$. The learning rate ratio $\alpha = 500$ is tuned on a specific image (i.e., "F16") under a specific corruption level (i.e., $p = 50\%$) for achieving the highest PSNR, and is subsequently fixed for all experiments. Finally, our method is *untrained* (i.e., it does *not* require any training data), so for fair comparisons we do not compare with trained image denoising methods such as [40–42].

**No need to tune model width or terminate early.** We compare our method with a variant of DIP that we call DIP-$\ell_1$, which is based on solving $\min_{\boldsymbol{\theta}} \ell(\phi(\boldsymbol{\theta}) - \mathbf{y})$ with $\ell(\cdot)$ being $\|\cdot\|_1$. As shown in Figure 1b, DIP-$\ell_1$ requires either choosing appropriate network width or early termination to

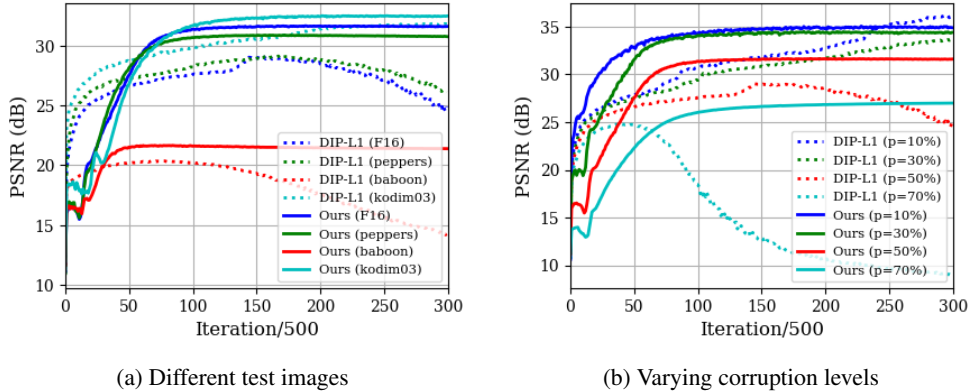

(a) Different test images　　　　　　　　(b) Varying corruption levels

Figure 4: **Learning curves for robust image recovery with different test images (a) and varying corruption levels (b).** DIP-$\ell_1$ requires case-dependent early stopping while DOP (our method) does not.

avoid overfitting. Note that neither of these can be carried out in practice without the true (clean) image. On the other hand, our method does not require tuning network width or early termination. Its performance continues to improve as training proceeds until stabilises.

**Handling different images and varying corruption levels.** The benefit mentioned above enables our method to handle different images and varying corruption levels *without* the need to tune network width, termination condition and any other learning parameters. In our experiments, we fix the number of channels in each convolutional layer to be 128, run 150,000 gradient descent iterations and display the output image. The results are shown in Figure 3 (for four different test images) and Figure 5 (for four corruption levels, see Appendix C). For DIP and DIP-$\ell_1$, we report the result with the highest PSNR in the learning process (i.e., we perform the best early termination for evaluation purposes – note that this cannot be carried out in practice). Despite that, our method obtains better recovery quality in terms of PSNR for all cases. We display the learning curves for these experiments in Figure 4a and Figure 4b, which show that our method does not overfit for all cases while DIP-$\ell_1$ requires a case-dependent early termination.

## 5 Conclusion

In this work, we have shown both theoretically and empirically that the benefits of implicit bias of gradient descent can be extended to over-parameterization of two low-dimensional structures. The key to the success is the choice of discrepant learning rates that can properly regulate the optimization path so that it converges to the desired optimal solution. Such a framework frees us from the need of prior knowledge in the structures or from the lack of scalability of previous approaches. This has led to state of the art recovery results for both low-rank matrices and natural images. We hope this work may encourage people to investigate in the future if the same framework and idea generalize to a mixture of multiple and broader families of structures.

## Broader Impact

Robust learning of structured signals from high-dimensional data has a wide range of applications, including imaging processing, computer vision, recommender systems, generative models and many more. In this work, we presented a new type of practical methods and provided improved understandings of solving these problems via over-parameterized models. In particular, our method exploits the implicit bias introduced by the learning algorithm, with the underlying driving force being the intrinsic structure of the data itself rather than human handcrafting. Such a design methodology helps to eliminate human bias in the design process, hence provides the basis for developing truly fair machine learning systems.

## Acknowledgments and Disclosure of Funding

CY and YM acknowledge support from Tsinghua-Berkeley Shenzhen Institute (TBSI) Research Fund. YM acknowledges support from ONR grant N00014-20-1-2002 and the joint Simons

Foundation-NSF DMS grant #2031899, as well as support from Berkeley AI Research (BAIR), Berkeley FHL Vive Center for Enhanced Reality, and Berkeley Center for Augmented Cognition. ZZ acknowledges support from NSF grant #2008460. QQ acknowledges support from the Moore-Sloan fellowship and NSF DMS #2009752. We would like to thank Shuang Li for carefully proof-reading this manuscript, Yaodong Yu and Ryan Chan for the helpful discussions on connections between matrix factorization and deep image prior and on implicit bias with Hadamard product.

## Footnotes

[1]Nuclear norm (the tightest convex envelope to matrix rank) is defined as the sum of singular values.

[2]This is a commonly used approach for robust optimization problems, such as robust regression [24], robust subspace learning [25–27], robust phase retrieval [17, 28], robust matrix recovery [16, 19], and many more.

[3]By a lifting trick such as [46, 47], our method can be extended to handling arbitrary rectangular matrices.

[4]Our result also differs from [53–55] which analyze the effect of initial large learning rates.

[5]Here, $H, W$ are height and width of the image, respectively. $C$ denotes the number of channels of the image, where $C = 1$ for a greyscale image and $C = 3$ for a color image with RGB channels.

[6]Any $\mathcal{A} : \mathbb{R}^{n \times n} \to \mathbb{R}^m$ can be written as $\mathcal{A}(\mathbf{Z}) = [\langle \mathbf{A}_1, \mathbf{Z} \rangle, \dots, \langle \mathbf{A}_m, \mathbf{Z} \rangle]$ for some $\{\mathbf{A}_i \in \mathbb{R}^{n \times n}\}_{i=1}^m$.

[7]http://people.eecs.berkeley.edu/~yima/matrix-rank/sample_code.html

[8]http://www.cs.tut.fi/~foi/GCF-BM3D/index.html#ref_results

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
