[Supplementary Material]

# Robust Recovery via Implicit Bias of Discrepant Learning Rates for Double Over-parameterization

## —*Supplementary Materials*—

## A    Implicit Bias of Discrepant Learning Rates in Linear Regression

In this part of the appendix, let us use a classical result for *underdetermined* linear regression problem to build up some intuitions behind the implicit bias of gradient descent for our problem formulation of robust learning problems. The high level message we aim to deliver through the simple example is that

- Gradient descent implicitly biases towards solutions with minimum $\ell_2$-norm.
- Discrepant learning rates lead to solutions with minimum *weighted* $\ell_2$-norm.

**Underdetermined linear regression.**    Given observation $\mathbf{b} \in \mathbb{R}^{n_1}$ and wide data matrix $\mathbf{W} \in \mathbb{R}^{n_1 \times n_2}$ $(n_2 > n_1)$, we want to find $\boldsymbol{\theta}$ which is a solution to

$$\min_{\boldsymbol{\theta} \in \mathbb{R}^{n_2}} \varphi(\boldsymbol{\theta}) \;=\; \frac{1}{2} \|\mathbf{b} - \mathbf{W}\boldsymbol{\theta}\|_2^2. \tag{16}$$

For $n_2 > n_1$ and full row-rank $\mathbf{W}$, the underdetermined problem (16) obviously has *infinite* many solutions, which forms a set

$$\mathcal{S} \;:=\; \left\{ \; \boldsymbol{\theta}_{ln} + \mathbf{n} \;\mid\; \boldsymbol{\theta}_{ln} = \mathbf{W}^{\dagger}\mathbf{b}, \quad \mathbf{n} \in \mathcal{N}(\mathbf{W}) \; \right\},$$

where $\mathbf{W}^{\dagger} := \mathbf{W}^{\top} \left( \mathbf{W}\mathbf{W}^{\top} \right)^{-1}$ denotes the pseudo-inverse of $\mathbf{W}$, and

$$\mathcal{N}(\mathbf{W}) := \{ \mathbf{n} \mid \mathbf{W}\mathbf{n} = \mathbf{0} \}, \quad \mathcal{R}(\mathbf{W}) := \left\{ \mathbf{z} \mid \mathbf{z} = \mathbf{W}^{\top}\mathbf{v} \right\}$$

are the null space and row space of $\mathbf{W}$, respectively. Simple derivation shows that $\boldsymbol{\theta}_{ln}$ is a particular *least $\ell_2$-norm* solution to (16), that minimizes

$$\min_{\boldsymbol{\theta} \in \mathbb{R}^{n_2}} \; \frac{1}{2} \|\boldsymbol{\theta}\|_2^2, \quad \text{s.t.} \quad \mathbf{W}\boldsymbol{\theta} \;=\; \mathbf{b}.$$

**Gradient descent biases towards $\boldsymbol{\theta}_{ln}$.**    Starting from any initialization $\boldsymbol{\theta}_0$, gradient descent

$$\boldsymbol{\theta}_{k+1} \;=\; \boldsymbol{\theta}_k - \tau_{ls} \cdot \mathbf{W}^{\top} \left( \mathbf{W}\boldsymbol{\theta}_k - \mathbf{b} \right) \tag{17}$$

with a sufficiently small learning rate[9] $\tau_{ls}$ always finds one of the global solutions for (16). Furthermore, it is now well-understood [59] that whenever the initialization $\boldsymbol{\theta}_0$ has zero component in $\mathcal{N}(\mathbf{W})$ (i.e., $\mathcal{P}_{\mathcal{N}(\mathbf{W})}(\boldsymbol{\theta}_0) = \mathbf{0}$), one interesting phenomenon is that the iterates $\boldsymbol{\theta}_{\infty}$ in (17) implicitly bias towards the minimum $\ell^2$-norm solution $\boldsymbol{\theta}_{ln}$. This happens because once initialized in $\mathcal{R}(\mathbf{W})$, gradient descent (17) implicitly biases towards iterates staying within $\mathcal{R}(\mathbf{W})$, such that

$$\mathcal{P}_{\mathcal{R}(\mathbf{W})}(\boldsymbol{\theta}_{\infty}) \;=\; \boldsymbol{\theta}_{ln}, \quad \mathcal{P}_{\mathcal{N}(\mathbf{W})}(\boldsymbol{\theta}_{\infty}) \;=\; \mathcal{P}_{\mathcal{N}(\mathbf{W})}(\boldsymbol{\theta}_0) = \mathbf{0}.$$

As we can see, a particular algorithm enforces specific regularization on the final solution.

**Implicit bias of discrepant learning rates.**    The gradient update in (17) uses the same learning rate $\tau_{ls}$ for all coordinates of $\boldsymbol{\theta}$. If we use different learning rates for each coordinate (i.e., $\boldsymbol{\Lambda}$ is a diagonal matrix with positive diagonals)

$$\boldsymbol{\theta}_{k+1} \;=\; \boldsymbol{\theta}_k - \tau_{ls} \cdot \boldsymbol{\Lambda} \cdot \mathbf{W}^{\top} \left( \mathbf{W}\boldsymbol{\theta}_k - \mathbf{b} \right), \tag{18}$$

then by following a similar argument we conclude that the gradient update in (18) converges to a weighted regularized solution for

$$\min_{\boldsymbol{\theta} \in \mathbb{R}^{n_2}} \; \frac{1}{2} \left\| \boldsymbol{\Lambda}^{-1/2} \boldsymbol{\theta} \right\|_2^2, \quad \text{s.t.} \quad \mathbf{W}\boldsymbol{\theta} \;=\; \mathbf{b}. \tag{19}$$

*Remark* 1. Let $\sigma_i$ be the $i$-th diagonal of $\mathbf{\Lambda}$ and $\theta_i$ be the $i$-th element of $\boldsymbol{\theta}$. Then in (18), $\sigma_i \tau_{ls}$ is the learning rate for the variable $\theta_i$, which varies for different variables. In words, the relation between (18) and (19) implies that for one particular optimization variable (e.g., $\theta_i$) a large learning rate $\sigma_i \tau_{ls}$ in (18) leads to a small implicit regularization effect in (19). From a high-level perspective, this happens because a larger learning rate allows the optimization variable to move faster away from its initial point, resulting in a weaker regularization effect (which penalizes the distance of the variable to the initialization) on its solution path.

An alternative explanation of this is through a change of variable $\boldsymbol{\theta} = \mathbf{\Lambda}^{1/2}\widetilde{\boldsymbol{\theta}}$. Suppose we minimize

$$\min_{\widetilde{\boldsymbol{\theta}} \in \mathbb{R}^{n_2}} \widetilde{\varphi}(\widetilde{\boldsymbol{\theta}}) \;=\; \frac{1}{2} \left\| \mathbf{b} - \mathbf{W}\mathbf{\Lambda}^{1/2}\widetilde{\boldsymbol{\theta}} \right\|_2^2 \tag{20}$$

via standard gradient descent with a single learning rate

$$\widetilde{\boldsymbol{\theta}}_{k+1} \;=\; \widetilde{\boldsymbol{\theta}}_k - \tau_{ls} \cdot \mathbf{\Lambda}^{1/2} \cdot \mathbf{W}^{\top} \left( \mathbf{W}\mathbf{\Lambda}^{1/2}\widetilde{\boldsymbol{\theta}}_k - \mathbf{b} \right). \tag{21}$$

Thus, once initialized in $\mathcal{R}(\mathbf{W}\mathbf{\Lambda}^{1/2})$, gradient descent (21) converges to the least $\ell_2$-norm solution to (20), i.e., the solution of the following problem

$$\min_{\widetilde{\boldsymbol{\theta}} \in \mathbb{R}^{n_2}} \frac{1}{2} \left\| \widetilde{\boldsymbol{\theta}} \right\|_2^2, \quad \text{s.t.} \quad \mathbf{W}\mathbf{\Lambda}^{1/2}\widetilde{\boldsymbol{\theta}} \;=\; \mathbf{b}. \tag{22}$$

Finally, plugging $\widetilde{\boldsymbol{\theta}} = \mathbf{\Lambda}^{-1/2}\boldsymbol{\theta}$ into (21) and (22) gives (18) and (19), respectively, also indicating the gradient update (18) induces implicit weighted regularization towards the solution of (19).

# B Proof of Theorem 1

In this part of the appendix, we provide the proof to our main technical result (i.e., Theorem 1) in Section 3. To make this part self-contained, we restate our result as follows.

*Theorem* 2. *Assume that the measurement matrices* $\mathbf{A}_1, \mathbf{A}_2, \ldots, \mathbf{A}_m$ *are symmetric and commutable, i.e.*

$$\mathbf{A}_i \mathbf{A}_j \;=\; \mathbf{A}_j \mathbf{A}_i, \quad \forall\, 1 \le i \ne j \le m,$$

*and the gradient flows of* $\mathbf{U}_t(\gamma)$, $\mathbf{g}_t(\gamma)$, *and* $\mathbf{h}_t(\gamma)$ *satisfy*

$$\dot{\mathbf{U}}_t(\gamma) \;=\; \lim_{\tau \to 0} \frac{\mathbf{U}_{t+\tau}(\gamma) - \mathbf{U}_t(\gamma)}{\tau} \;=\; -\mathcal{A}^* \left( \mathbf{r}_t(\gamma) \right) \mathbf{U}_t(\gamma), \tag{23}$$

$$\begin{bmatrix} \dot{\mathbf{g}}_t(\gamma) \\ \dot{\mathbf{h}}_t(\gamma) \end{bmatrix} \;=\; \lim_{\tau \to 0} \left( \begin{bmatrix} \mathbf{g}_{t+\tau}(\gamma) \\ \mathbf{h}_{t+\tau}(\gamma) \end{bmatrix} - \begin{bmatrix} \mathbf{g}_t(\gamma) \\ \mathbf{h}_t(\gamma) \end{bmatrix} \right) / \tau \;=\; - \alpha \cdot \begin{bmatrix} \mathbf{r}_t(\gamma) \circ \mathbf{g}_t(\gamma) \\ -\mathbf{r}_t(\gamma) \circ \mathbf{h}_t(\gamma), \end{bmatrix}, \tag{24}$$

*with* $\mathbf{r}_t(\gamma) \;=\; \mathcal{A}(\mathbf{U}_t(\gamma)\mathbf{U}_t^{\top}(\gamma)) + \mathbf{g}_t(\gamma) \circ \mathbf{g}_t(\gamma) - \mathbf{h}_t(\gamma) \circ \mathbf{h}_t(\gamma) - \mathbf{y}$, *and they are initialized by*

$$\mathbf{U}_0(\gamma) \;=\; \gamma \mathbf{I}, \quad \mathbf{g}_0(\gamma) \;=\; \gamma \mathbf{1}, \quad h_0(\gamma) \;=\; \gamma \mathbf{1}.$$

*Let* $\mathbf{X}_t(\gamma) = \mathbf{U}_t(\gamma)\mathbf{U}_t^{\top}(\gamma)$, *and let* $\mathbf{X}_\infty(\gamma)$, $\mathbf{g}_\infty(\gamma)$, *and* $\mathbf{g}_\infty(\gamma)$ *be the limit points defined as*

$$\mathbf{X}_\infty(\gamma) \;:=\; \lim_{t \to +\infty} \mathbf{X}_t(\gamma), \quad \mathbf{g}_\infty(\gamma) \;:=\; \lim_{t \to +\infty} \mathbf{g}_t(\gamma), \quad \mathbf{h}_\infty(\gamma) \;:=\; \lim_{t \to +\infty} \mathbf{h}_t(\gamma). \tag{25}$$

*Suppose that our initialization is infinitesimally small such that*

$$\widehat{\mathbf{X}} \;:=\; \lim_{\gamma \to 0} \mathbf{X}_\infty(\gamma), \quad \widehat{\mathbf{g}} \;:=\; \lim_{\gamma \to 0} \mathbf{g}_\infty(\gamma), \quad \widehat{\mathbf{h}} \;:=\; \lim_{\gamma \to 0} \mathbf{h}_\infty(\gamma)$$

*exist and* $(\widehat{\mathbf{X}}, \widehat{\mathbf{g}}, \widehat{\mathbf{h}})$ *is a global optimal solution to*

$$\min_{\mathbf{U} \in \mathbb{R}^{n \times r'}, \{\mathbf{g}, \mathbf{h}\} \subseteq \mathbb{R}^m} f(\mathbf{U}, \mathbf{g}, \mathbf{h}) \;:=\; \frac{1}{4} \left\| \mathcal{A}\left( \mathbf{U}\mathbf{U}^{\top} \right) + (\mathbf{g} \circ \mathbf{g} - \mathbf{h} \circ \mathbf{h}) - \mathbf{y} \right\|_2^2, \tag{26}$$

*with* $\mathcal{A}(\widehat{\mathbf{X}}) + \widehat{\mathbf{s}} \;=\; \mathbf{y}$ *and* $\widehat{\mathbf{s}} \;=\; \widehat{\mathbf{g}} \circ \widehat{\mathbf{g}} - \widehat{\mathbf{h}} \circ \widehat{\mathbf{h}}$. *Then we have* $\widehat{\mathbf{g}} \circ \widehat{\mathbf{h}} = \mathbf{0}$, *and* $(\widehat{\mathbf{X}}, \widehat{\mathbf{s}})$ *is also a global optimal solution to*

$$\min_{\mathbf{X} \in \mathbb{R}^{n \times n}, \mathbf{s} \in \mathbb{R}^m} \| \mathbf{X} \|_* \;+\; \lambda \cdot \| \mathbf{s} \|_1, \quad \text{s.t.} \quad \mathcal{A}(\mathbf{X}) + \mathbf{s} \;=\; \mathbf{y}, \;\; \mathbf{X} \succeq \mathbf{0}. \tag{27}$$

*with* $\lambda = \alpha^{-1}$ *and* $\alpha > 0$ *being the balancing parameter in* (24).

*Proof.* From (23), we can derive the gradient flow for $\mathbf{X}_t(\gamma)$ via chain rule

$$\dot{\mathbf{X}}_t(\gamma) \;=\; \dot{\mathbf{U}}_t(\gamma)\mathbf{U}_t^\top(\gamma) \;+\; \mathbf{U}_t(\gamma)\dot{\mathbf{U}}_t^\top(\gamma) \;=\; -\mathcal{A}^*(\mathbf{r}_t(\gamma))\mathbf{X}_t(\gamma) - \mathbf{X}_t(\gamma)\mathcal{A}^*(\mathbf{r}_t(\gamma)). \quad (28)$$

We want to show that when the initialization is infinitesimally small (i.e., $\gamma \to 0$), the limit points of the gradient flows $\mathbf{X}_t(\gamma) = \mathbf{U}_t(\gamma)\mathbf{U}_t^\top(\gamma)$ and $\mathbf{s}_t(\gamma) = \mathbf{g}_t(\gamma) \circ \mathbf{g}_t(\gamma) - \mathbf{h}_t(\gamma) \circ \mathbf{h}_t(\gamma)$ are optimal solutions for (27) as $t \to +\infty$. Towards this goal, let us first look at the optimality condition for (27). From Lemma 1, we know that if $(\widehat{\mathbf{X}}, \widehat{\mathbf{s}})$ with

$$\widehat{\mathbf{X}} \;=\; \lim_{\gamma \to 0} \mathbf{X}_\infty(\gamma), \quad \widehat{\mathbf{s}} \;=\; \widehat{\mathbf{g}} \circ \widehat{\mathbf{g}} - \widehat{\mathbf{h}} \circ \widehat{\mathbf{h}} \quad \text{with} \quad \widehat{\mathbf{g}} \;:=\; \lim_{\gamma \to 0} \mathbf{g}_\infty(\gamma), \quad \widehat{\mathbf{h}} \;:=\; \lim_{\gamma \to 0} \mathbf{h}_\infty(\gamma)$$

is an optimal solution for (27) then there exists a *dual certificate* $\boldsymbol{\nu}$ such that

$$\mathcal{A}(\widehat{\mathbf{X}}) + \widehat{\mathbf{s}} \;=\; \mathbf{y}, \quad (\mathbf{I} - \mathcal{A}^*(\boldsymbol{\nu})) \cdot \widehat{\mathbf{X}} \;=\; \mathbf{0}, \quad \mathcal{A}^*(\boldsymbol{\nu}) \preceq \mathbf{I}, \quad \boldsymbol{\nu} \in \lambda \cdot \mathrm{sign}(\widehat{\mathbf{s}}), \quad \widehat{\mathbf{X}} \succeq \mathbf{0},$$

where $\mathrm{sign}(\widehat{\mathbf{s}})$ is defined in (32), which is the subdifferential of $\|\cdot\|_1$. Thus, it suffices to construct a dual certificate $\boldsymbol{\nu}$ such that $(\widehat{\mathbf{X}}, \widehat{\mathbf{s}})$ satisfies the equation above.

Since $(\widehat{\mathbf{X}}, \widehat{\mathbf{g}}, \widehat{\mathbf{h}})$ is a global optimal solution to (26), we automatically have $\mathcal{A}(\widehat{\mathbf{X}}) + \widehat{\mathbf{s}} \;=\; \mathbf{y}$ and $\widehat{\mathbf{X}} \succeq \mathbf{0}$. On the other hand, given that $\{\mathbf{A}_i\}_{i=1}^m$ commutes and (28) and (24) hold for $\mathbf{X}_t, \mathbf{g}_t$ and $\mathbf{h}_t$, by Lemma 2, we know that

$$\mathbf{X}_t(\gamma) \;=\; \exp\left(\mathcal{A}^*(\boldsymbol{\xi}_t(\gamma))\right) \cdot \mathbf{X}_0(\gamma) \cdot \exp\left(\mathcal{A}^*(\boldsymbol{\xi}_t(\gamma))\right), \quad (29)$$
$$\mathbf{g}_t(\gamma) \;=\; \mathbf{g}_0(\gamma) \circ \exp\left(\alpha\boldsymbol{\xi}_t(\gamma)\right), \quad \mathbf{h}_t(\gamma) \;=\; \mathbf{h}_0(\gamma) \circ \exp\left(-\alpha\boldsymbol{\xi}_t(\gamma)\right), \quad (30)$$

where $\boldsymbol{\xi}_t(\gamma) = -\int_0^t \mathbf{r}_\tau(\gamma)d\tau$. Let $\boldsymbol{\xi}_\infty(\gamma) := \lim_{t \to +\infty} \boldsymbol{\xi}_t(\gamma)$, by Lemma 3 and Lemma 4, we can construct

$$\boldsymbol{\nu}(\gamma) \;=\; \frac{\boldsymbol{\xi}_\infty(\gamma)}{\log\left(1/\gamma\right)},$$

such that

$$\lim_{\gamma \to 0} \mathcal{A}^*\left(\boldsymbol{\nu}(\gamma)\right) \;\preceq\; \mathbf{I}, \quad \lim_{\gamma \to 0} \left[\mathbf{I} - \mathcal{A}^*\left(\boldsymbol{\nu}(\gamma)\right)\right] \cdot \widehat{\mathbf{X}} \;=\; \mathbf{0},$$

and

$$\lim_{\gamma \to 0} \boldsymbol{\nu}(\gamma) \;\in\; \alpha^{-1} \cdot \mathrm{sign}(\widehat{\mathbf{s}}), \quad \lim_{\gamma \to 0} \mathbf{g}(\gamma) \circ \mathbf{h}(\gamma) \;=\; \mathbf{0}.$$

This shows the exists of the dual certificate $\boldsymbol{\nu}(\gamma)$ such that the optimality condition holds for $(\widehat{\mathbf{X}}, \widehat{\mathbf{s}})$. Hence, $(\widehat{\mathbf{X}}, \widehat{\mathbf{s}})$ is also a global optimal solution to (27). $\square$

**Lemma 1.** $(\widehat{\mathbf{X}}, \widehat{\mathbf{s}})$ *is an optimal solution for* (27) *if there exists a dual certificate* $\boldsymbol{\nu} \in \mathbb{R}^m$ *such that the following conditions hold:*

$$\mathcal{A}(\widehat{\mathbf{X}}) + \widehat{\mathbf{s}} \;=\; \mathbf{y}, \quad (\mathbf{I} - \mathcal{A}^*(\boldsymbol{\nu})) \cdot \widehat{\mathbf{X}} \;=\; \mathbf{0}, \quad \boldsymbol{\nu} \in \lambda \cdot \mathrm{sign}(\widehat{\mathbf{s}}), \quad \mathbf{I} \succeq \mathcal{A}^*(\boldsymbol{\nu}), \; \widehat{\mathbf{X}} \succeq \mathbf{0}, \quad (31)$$

*where* $\mathrm{sign}(\mathbf{s})$ *is the subdifferential of* $\|\mathbf{s}\|_1$ *with each entry*

$$\mathrm{sign}(s) \;:=\; \begin{cases} s/\left|s\right| & s \neq 0, \\ [-1, 1] & s = 0. \end{cases} \quad (32)$$

*Proof.* The Lagrangian function of the problem can be written as

$$\mathcal{L}\left(\mathbf{X}, \mathbf{s}, \boldsymbol{\nu}, \boldsymbol{\Gamma}\right) \;=\; \mathrm{trace}\left(\mathbf{X}\right) + \lambda\left\|\mathbf{s}\right\|_1 + \boldsymbol{\nu}^\top\left(\mathbf{y} - \mathcal{A}(\mathbf{X}) - \mathbf{s}\right) - \langle \mathbf{X}, \boldsymbol{\Gamma} \rangle,$$

with $\boldsymbol{\nu} \in \mathbb{R}^m$ and $\boldsymbol{\Gamma} \in \mathbb{R}^{n \times n}$ being the dual variables, where $\boldsymbol{\Gamma} \succeq \mathbf{0}$. Thus, we can derive the Karush-Kuhn-Tucker (KKT) optimality condition for (27) as

$$\mathbf{0} \in \partial\mathcal{L}: \quad \mathbf{I} - \boldsymbol{\Gamma} - \mathcal{A}^*(\boldsymbol{\nu}) \;=\; \mathbf{0}, \quad \boldsymbol{\nu} \in \lambda \cdot \partial\left\|\mathbf{s}\right\|_1 = \lambda \cdot \mathrm{sign}\left(\mathbf{s}\right),$$
$$\text{feasibility}: \quad \mathcal{A}(\mathbf{X}) + \mathbf{s} \;=\; \mathbf{y}, \; \mathbf{X} \succeq \mathbf{0}, \; \boldsymbol{\Gamma} \succeq \mathbf{0},$$
$$\text{complementary slackness}: \quad \boldsymbol{\Gamma} \cdot \mathbf{X} \;=\; \mathbf{0},$$

where $\partial(\cdot)$ denotes the subdifferential operator and $\text{sign}(\mathbf{s})$ is the subdifferential of $\|\mathbf{s}\|_1$ with each entry

$$\text{sign}(s) = \begin{cases} s/|s| & s \neq 0, \\ [-1, 1] & s = 0. \end{cases}$$

Thus, we know that $\left(\widehat{\mathbf{X}}, \widehat{\mathbf{s}}\right)$ is global solution to (27) as long as there exists a *dual certificate* $\boldsymbol{\nu}$ such that (31) holds, where we eliminated $\boldsymbol{\Gamma}$ by plugging in $\boldsymbol{\Gamma} = \mathbf{I} - \mathcal{A}^*(\boldsymbol{\nu})$. $\qquad\square$

*Lemma* 2. *Suppose that* $\{\mathbf{A}_i\}_{i=1}^m$ *commutes. Suppose* (14) *and* (24) *hold for* $\mathbf{X}_t$, $\mathbf{g}_t$ *and* $\mathbf{h}_t$, *then*

$$\mathbf{X}_t = \exp\left(\mathcal{A}^*(\boldsymbol{\xi}_t)\right) \cdot \mathbf{X}_0 \cdot \exp\left(\mathcal{A}^*(\boldsymbol{\xi}_t)\right) \tag{33}$$
$$\mathbf{g}_t = \mathbf{g}_0 \circ \exp\left(\alpha\boldsymbol{\xi}_t\right), \quad \mathbf{h}_t = \mathbf{h}_0 \circ \exp\left(-\alpha\boldsymbol{\xi}_t\right), \tag{34}$$

*where* $\boldsymbol{\xi}_t = -\int_0^t \mathbf{r}_\tau d\tau$.

*Proof.* From (24), we know that

$$\frac{d\mathbf{g}_t}{dt} = -\alpha\mathbf{r}_t \circ \mathbf{g}_t,$$

where the differentiation $\frac{d\mathbf{g}_t}{dt}$ is entrywise for $\mathbf{g}_t$. Thus, we have

$$\int_0^t \frac{d\mathbf{g}_\tau}{\mathbf{g}_\tau} = -\alpha \int_0^t \mathbf{r}_\tau d\tau \implies \log\mathbf{g}_t - \log\mathbf{g}_0 = \alpha\boldsymbol{\xi}_t \implies \mathbf{g}_t = \mathbf{g}_0 \circ \exp\left(\alpha\boldsymbol{\xi}_t\right),$$

where all the operators are entrywise. Similarly, $\mathbf{h}_t = \mathbf{h}_0 \circ \exp\left(-\alpha\boldsymbol{\xi}_t\right)$ holds.

For (33), by using (28) and the fact that $\{\mathbf{A}_i\}_{i=1}^m$ commutes, we can derive it with an analogous argument. $\qquad\square$

*Lemma* 3. *Under the settings of Theorem 2 and Lemma 2, for any* $\gamma > 0$ *there exists*

$$\boldsymbol{\nu}(\gamma) = \frac{\boldsymbol{\xi}_\infty(\gamma)}{\log(1/\gamma)}, \tag{35}$$

*such that*

$$\lim_{\gamma \to 0} \mathcal{A}^*\left(\boldsymbol{\nu}(\gamma)\right) \preceq \mathbf{I}, \quad \lim_{\gamma \to 0}\left[\mathbf{I} - \mathcal{A}^*\left(\boldsymbol{\nu}(\gamma)\right)\right] \cdot \widehat{\mathbf{X}} = \mathbf{0},$$

*where* $\boldsymbol{\xi}_\infty(\gamma) = \lim_{t \to 0} \boldsymbol{\xi}_t(\gamma)$ *with* $\boldsymbol{\xi}_t(\gamma) = -\int_0^t \mathbf{r}_\tau(\gamma)d\tau$.

*Proof.* Given $\mathbf{U}_0 = \gamma\mathbf{I}$, we have $\mathbf{X}_0 = \mathbf{U}_0\mathbf{U}_0^\top = \gamma^2\mathbf{I}$. By the expression for $\mathbf{X}_t$ in (29), we have

$$\mathbf{X}_\infty(\gamma) = \gamma^2 \cdot \exp\left(2\mathcal{A}^*\left(\boldsymbol{\xi}_\infty(\gamma)\right)\right) \tag{36}$$

where $\boldsymbol{\xi}_\infty(\gamma) = \lim_{t \to \infty} \boldsymbol{\xi}_t(\gamma)$. Because $\{\mathbf{A}_i\}_{i=1}^m$ are symmetric and they commute, we know that they are simultaneously diagonalizable by an orthonormal basis $\boldsymbol{\Omega} = [\boldsymbol{\omega}_1, \dots, \boldsymbol{\omega}_n] \in \mathbb{R}^{n \times n}$, i.e.,

$$\boldsymbol{\Omega}\mathbf{A}_i\boldsymbol{\Omega}^\top = \boldsymbol{\Lambda}_i, \quad \boldsymbol{\Lambda}_i \text{ diagonal}, \quad \forall\, i = 1, 2, \dots, m,$$

and so is $\mathcal{A}^*(\mathbf{b})$ for any $\mathbf{b} \in \mathbb{R}^m$. Therefore, we have

$$\lambda_k\left(\mathbf{X}_\infty(\gamma)\right) = \gamma^2 \cdot \exp\left(2\lambda_k\left(\mathcal{A}^*\left(\boldsymbol{\xi}_\infty(\gamma)\right)\right)\right) = \exp\left(2\lambda_k\left(\mathcal{A}^*\left(\boldsymbol{\xi}_\infty(\gamma)\right)\right) + 2\log\gamma\right), \tag{37}$$

where $\lambda_k(\cdot)$ denotes the $k$-th eigenvalue with respect to the $k$-th basis $\boldsymbol{\omega}_k$. Moreover $\mathbf{X}_\infty(\gamma)$ and its limit $\widehat{\mathbf{X}}$ have the same eigen-basis $\boldsymbol{\Omega}$. Since we have $\mathbf{X}_\infty(\gamma)$ converges to $\widehat{\mathbf{X}}$ when $\gamma \to 0$, then we have the eigenvalues

$$\lambda_k\left(\mathbf{X}_\infty(\gamma)\right) \to \lambda_k(\widehat{\mathbf{X}}), \quad \forall\, k = 1, 2, \dots, n, \tag{38}$$

whenever $\gamma \to 0$.

**Case 1:** $\lambda_k(\widehat{\mathbf{X}}) > 0$. For any $k$ such that $\lambda_k(\widehat{\mathbf{X}}) > 0$, from (37) and (38), we have

$$\exp\left(2\lambda_k\left(\mathcal{A}^*\left(\boldsymbol{\xi}_\infty(\gamma)\right)\right) + 2\log\gamma\right) \;\to\; \lambda_k(\widehat{\mathbf{X}}),$$

so that

$$2\lambda_k\left(\mathcal{A}^*\left(\boldsymbol{\xi}_\infty(\gamma)\right)\right) + 2\log\gamma - \log\lambda_k(\widehat{\mathbf{X}}) \;\to\; 0,$$

which further implies that

$$\lambda_k\left(\mathcal{A}^*\left(\frac{\boldsymbol{\xi}_\infty(\gamma)}{\log(1/\gamma)}\right)\right) - 1 - \frac{\log\lambda_k(\widehat{\mathbf{X}})}{2\log(1/\gamma)} \;\to\; 0.$$

Now if we construct $\boldsymbol{\nu}(\gamma)$ as (35), so that we conclude

$$\lim_{\gamma\to 0}\lambda_k\left(\mathcal{A}^*(\boldsymbol{\nu}(\gamma))\right) \;=\; 1, \tag{39}$$

for any $k$ such that $\lambda_k(\widehat{\mathbf{X}}) > 0$.

**Case 2:** $\lambda_k(\widehat{\mathbf{X}}) = 0$. On the other hand, for any $k$ such that $\lambda_k(\widehat{\mathbf{X}}) = 0$, similarly from (37) and (38), we have

$$\exp\left(2\lambda_k\left(\mathcal{A}^*\left(\boldsymbol{\xi}_\infty(\gamma)\right)\right) + 2\log\gamma\right) \;\to\; 0,$$

when $\gamma \to 0$. Thus, for any small $\epsilon \in (0,1)$, there exists some $\gamma_0 \in (0,1)$ such that for all $\gamma < \gamma_0$,

$$\exp\left(2\lambda_k\left(\mathcal{A}^*\left(\boldsymbol{\xi}_\infty(\gamma)\right)\right) + 2\log\gamma\right) \;<\; \epsilon,$$

which implies that

$$\lambda_k\left(\mathcal{A}^*\left(\frac{\boldsymbol{\xi}_\infty(\gamma)}{\log(1/\gamma)}\right)\right) - 1 \;<\; \frac{\log\epsilon}{2\log(1/\gamma)} \;<\; 0.$$

Thus, given the construction of $\boldsymbol{\nu}(\gamma)$ in (35), we have

$$\lambda_k\left(\mathcal{A}^*(\boldsymbol{\nu}(\gamma))\right) \;<\; 1, \quad \forall\,\gamma \;<\; \gamma_0,$$

which further implies that for any $k$ with $\lambda_k(\widehat{\mathbf{X}}) = 0$, we have

$$\lim_{\gamma\to 0}\lambda_k\left(\mathcal{A}^*(\boldsymbol{\nu}(\gamma))\right) \;<\; 1. \tag{40}$$

**Putting things together.** Combining our analysis in (39) and (40), we obtain

$$\lim_{\gamma\to 0}\mathcal{A}^*(\boldsymbol{\nu}(\gamma)) \;\preceq\; \mathbf{I}.$$

On the other hand, per our analysis, we know that there exists an orthogonal matrix $\boldsymbol{\Omega} \in \mathbb{R}^{n\times n}$, such that $\mathcal{A}^*(\boldsymbol{\nu}(\gamma))$ and $\widehat{\mathbf{X}}$ can be simultaneously diagonalized. Thus, we have

$$[\mathbf{I} - \mathcal{A}^*(\boldsymbol{\nu}(\gamma))] \cdot \widehat{\mathbf{X}} \;=\; \boldsymbol{\Omega} \cdot \left(\mathbf{I} - \boldsymbol{\Lambda}_{\mathcal{A}^*(\boldsymbol{\nu}(\gamma))}\right) \cdot \boldsymbol{\Lambda}_{\widehat{\mathbf{X}}} \cdot \boldsymbol{\Omega}^\top,$$

where $\boldsymbol{\Lambda}_{\mathcal{A}^*(\boldsymbol{\nu}(\gamma))}$ and $\boldsymbol{\Lambda}_{\widehat{\mathbf{X}}}$ are diagonal matrices, with entries being the eigenvalues of $\mathcal{A}^*(\boldsymbol{\nu}(\gamma))$ and $\boldsymbol{\Lambda}_{\widehat{\mathbf{X}}}$, respectively. From our analysis for Case 1 and Case 2, we know that $\lim_{\gamma\to 0}\left(\mathbf{I} - \boldsymbol{\Lambda}_{\mathcal{A}^*(\boldsymbol{\nu}(\gamma))}\right) \cdot \boldsymbol{\Lambda}_{\widehat{\mathbf{X}}} = \mathbf{0}$. Therefore, we also have

$$\lim_{\gamma\to 0}\;[\mathbf{I} - \mathcal{A}^*(\boldsymbol{\nu}(\gamma))] \cdot \widehat{\mathbf{X}} \;=\; \mathbf{0},$$

as desired. $\qquad\square$

*Lemma* 4. *Under the settings of Theorem 2 and Lemma 2, for any $\gamma > 0$ there exists*

$$\boldsymbol{\nu}(\gamma) \;=\; \frac{\boldsymbol{\xi}_\infty(\gamma)}{\log(1/\gamma)}, \tag{41}$$

*such that*

$$\lim_{\gamma\to 0}\boldsymbol{\nu}(\gamma) \;\in\; \alpha^{-1}\cdot\mathrm{sign}(\widehat{\mathbf{s}}), \quad \lim_{\gamma\to 0}\mathbf{g}(\gamma)\circ\mathbf{h}(\gamma) \;=\; \mathbf{0}, \tag{42}$$

*where $\widehat{\mathbf{s}} = \widehat{\mathbf{g}}\circ\widehat{\mathbf{g}} - \widehat{\mathbf{h}}\circ\widehat{\mathbf{h}}$, and $\boldsymbol{\xi}_\infty(\gamma) = \lim_{t\to 0}\boldsymbol{\xi}_t(\gamma)$ with $\boldsymbol{\xi}_t(\gamma) = -\int_0^t \mathbf{r}_\tau(\gamma)d\tau$.*

*Proof.* Let $g^i_\infty(\gamma)$ and $h^i_\infty(\gamma)$ be the $i$th coordinate of $\mathbf{g}_\infty(\gamma)$ and $\mathbf{h}_\infty(\gamma)$ defined in (25), respectively. It follows from (30) that

$$g^i_\infty(\gamma) \;=\; \gamma \cdot \exp\left(\alpha \cdot \xi^i_\infty(\gamma)\right), \quad h^i_\infty(\gamma) \;=\; \gamma \cdot \exp\left(-\alpha \cdot \xi^i_\infty(\gamma)\right), \quad \forall\, i \;=\; 1, 2, \ldots, m.$$

When $\gamma \to 0$, we have

$$g^i_\infty(\gamma) \cdot h^i_\infty(\gamma) \;=\; \gamma^2 \;\to\; 0, \quad \forall\, i \;=\; 1, 2, \ldots, m,$$

so that $\lim_{\gamma \to 0} \mathbf{g}(\gamma) \circ \mathbf{h}(\gamma) \;=\; \mathbf{0}$. This also implies that either $g^i_\infty(\gamma)$ or $h^i_\infty(\gamma)$ for any $i = 1, 2, \ldots, m$.

On the other hand, let us define

$$\mathbf{s}_\infty(\gamma) \;=\; \mathbf{g}_\infty(\gamma) \circ \mathbf{g}_\infty(\gamma) \;-\; \mathbf{h}_\infty(\gamma) \circ \mathbf{h}_\infty(\gamma),$$

and let $s^i_\infty(\gamma)$ be the $i$th coordinate of $\mathbf{s}_\infty(\gamma)$ with

$$s^i_\infty(\gamma) \;=\; \gamma^2 \cdot \exp\left(2\alpha \cdot \xi^i_\infty(\gamma)\right) \;-\; \gamma^2 \cdot \exp\left(-2\alpha \cdot \xi^i_\infty(\gamma)\right). \tag{43}$$

Correspondingly, we know that $\widehat{\mathbf{s}} = \lim_{\gamma \to 0} \mathbf{s}_\infty(\gamma)$ and let $s^i$ be the $i$th coordinate of $\widehat{\mathbf{s}} = \widehat{\mathbf{g}} \circ \widehat{\mathbf{g}} - \widehat{\mathbf{h}} \circ \widehat{\mathbf{h}}$. In the following, we leverage on these to show that our construction of $\boldsymbol{\nu}(\gamma)$ satisfies (42). We classify the entries $\widehat{s}_i$ of $\widehat{\mathbf{s}}$ ($i = 1, 2, \ldots, m$) into three cases and analyze as follows.

- **Case 1: $\widehat{s}_i > 0$.** Since $\lim_{\gamma \to 0} s^i_\infty(\gamma) = \widehat{s}_i > 0$, from (43) we must have $\xi^i_\infty(\gamma) \to +\infty$ when $\gamma \to 0$, so that $\exp\left(2\alpha \cdot \xi^i_\infty(\gamma)\right) \to +\infty$ and $\exp\left(-2\alpha \cdot \xi^i_\infty(\gamma)\right) \to 0$. Therefore, when $\gamma \to 0$, we have

$$\gamma^2 \exp\left(2\alpha \cdot \xi^i_\infty(\gamma)\right) \;\to\; \widehat{s}_i \quad \Longrightarrow \quad 2\alpha \cdot \xi^i_\infty(\gamma) - 2\log\left(1/\gamma\right) - \log \widehat{s}_i \;\to\; 0,$$

$$\Longrightarrow \quad \nu_i(\gamma) \;=\; \frac{\xi^i_\infty(\gamma)}{\log\left(1/\gamma\right)} \;\to\; \frac{1}{\alpha}. \quad (\text{given } \log\left(1/\gamma\right) \to +\infty)$$

- **Case 2: $\widehat{s}_i < 0$.** Since $\lim_{\gamma \to 0} s^i_\infty(\gamma) = \widehat{s}_i < 0$, from (43) we must have $\xi^i_\infty(\gamma) \to -\infty$ when $\gamma \to 0$, so that $\exp\left(2\alpha \cdot \xi^i_\infty(\gamma)\right) \to 0$ and $\exp\left(-2\alpha \cdot \xi^i_\infty(\gamma)\right) \to +\infty$. Therefore, when $\gamma \to 0$, we have

$$-\gamma^2 \exp\left(-2\alpha \cdot \xi^i_\infty(\gamma)\right) \;\to\; \widehat{s}_i \quad \Longrightarrow \quad -2\alpha \cdot \xi^i_\infty(\gamma) + 2\log\left(1/\gamma\right) - \log \widehat{s}_i \;\to\; 0,$$

$$\Longrightarrow \quad \nu_i(\gamma) \;=\; \frac{\xi^i_\infty(\gamma)}{\log\left(1/\gamma\right)} \;\to\; -\frac{1}{\alpha}.$$

- **Case 3: $\widehat{s}_i = 0$.** Since $\lim_{\gamma \to 0} s^i_\infty(\gamma) = \widehat{s}_i = 0$, from (43) we must have $\gamma^2 \cdot \exp\left(2\alpha \cdot \xi^i_\infty(\gamma)\right) \to 0$ and $\gamma^2 \cdot \exp\left(-2\alpha \cdot \xi^i_\infty(\gamma)\right) \to 0$, when $\gamma \to 0$. Therefore, for any small $\epsilon \in (0, 1)$, there exists some $\gamma_0 > 0$, such that for all $\gamma \in (0, \gamma_0)$, we have

$$\gamma^2 \cdot \max\left\{\exp\left(2\alpha \cdot \xi^i_\infty(\gamma)\right), \; \exp\left(-2\alpha \cdot \xi^i_\infty(\gamma)\right)\right\} \;\leq\; \epsilon$$

$$\Longrightarrow \quad 2\alpha \cdot \max\left\{\frac{\xi^i_\infty(\gamma)}{\log\left(1/\gamma\right)}, -\frac{\xi^i_\infty(\gamma)}{\log\left(1/\gamma\right)}\right\} - 2 \;<\; \frac{\log \epsilon}{\log\left(1/\gamma\right)} \;<\; 0,$$

which further implies that

$$|\nu_i(\gamma)| \;=\; \max\left\{\frac{\xi^i_\infty(\gamma)}{\log\left(1/\gamma\right)}, -\frac{\xi^i_\infty(\gamma)}{\log\left(1/\gamma\right)}\right\} \;<\; \frac{1}{\alpha}.$$

Therefore, combining the results in the three cases above we obtain that

$$\lim_{\gamma \to 0} \nu_i(\gamma) \;=\; \frac{1}{\alpha} \operatorname{sign}(\widehat{s}_i) \;=\; \begin{cases} \frac{\widehat{s}_i}{\alpha |\widehat{s}_i|} & \widehat{s}_i \neq 0, \\ [-1/\alpha, 1/\alpha] & \widehat{s}_i = 0, \end{cases}$$

so that we have (42) holds. □

## C  Extra Experiments

Due to limited space in the main body, we here provide extra results for our experiments on robust image recovery presented in Section 4.2.

**Varying corruption levels.** In Figure 5, we display results of our method for robust image recovery with varying levels of salt-and-pepper corruption.

Figure 5: **Robust image recovery with** $10\%, 30\%, 50\%$, **and** $70\%$ **salt-and-pepper noise.** PSNR of the results is displayed below the images. For our method, all cases use the same network width, learning rate, and termination condition. For DIP and DIP-$\ell_1$, case-dependent early stopping is used which is essential for their good performance. Despite that, our method achieves the highest PSNRs and best visual quality.

## Footnotes

[9]This is because $\mathbf{W}\boldsymbol{\theta}_{k+1} - \mathbf{b} = \left( \mathbf{I} - \tau_{ls} \mathbf{W}\mathbf{W}^{\top} \right) \left( \mathbf{W}\boldsymbol{\theta}_k - \mathbf{b} \right)$. If we choose $\tau_{ls} < \left\| \mathbf{W}\mathbf{W}^{\top} \right\|^{-1}$, then $\|\mathbf{W}\boldsymbol{\theta}_k - \mathbf{b}\|$ converges to 0 geometrically.