[Reviews · NeurIPS 2020]

Review 1

Summary and Contributions: The paper proposes a low-rank and sparse matrix recovery method which considers a double over-parameterized model solved by gradient descent with discrepant learning rates. The proposed method is provably correct, free of prior information, and scalable. Numerical experiments including low-rank matrix recovery and image denoising are provided to justify the empirical effectiveness of the proposed approach. ==post-rebuttal== The authors have answered some of my questions in the rebuttal. Although I'm still concerned about its limitations in real applications, I decide to raise my score to 7 considering its theoretical contributions.

Strengths: The idea of applying double over-parameterization to robust low-rank matrix recovery has certain novelty, and the convergence analysis via dynamic gradient flow together with learning rate discussion is interesting and insightful to other related works.

Weaknesses: 1. Natural images especially non-texture type of images may not have intrinsic low-rank structures would limit the application of the proposed method on image denoising. 2. Since only the L1-regularization is imposed on the noise component, the proposed method can handle the salt-and-pepper noise well but the denoising performance is unknown for other types of noise. 3. The paper still needs more practical guidance on the learning rate selection case-by-case and a brief discussion of the impact of sampling rate on the performance.

Correctness: The proposed method and theoretical discussions are fine, but the numerical experiments have certain issues. The paper in fact only considers the image denoising case with the salt-and-pepper noise, without any other types of image degradation. So some statements about image recovery sound over-claimed. More numerical comparisons with state of the art should be included. Computational time comparison and convergence behavior discussions are missing as well.

Clarity: The paper is mostly well written but not always. For example, Section 2.1 is confusing in terms of notation and organization. The representation of s in (4) is not well explained/defined before using. In (5), the assumption about linear measurements b=As is not consistent with that in (4).

Relation to Prior Work: Yes.

Reproducibility: No

Additional Feedback: More implementation details could be added. In Figure 3, PSNR values should be put either under the subfigures or in the caption.


Review 2

Summary and Contributions: This paper studies recovery of positive semidefinite low rank matrices from undersampled linear measurements that are corrupted by sparse noise. The authors introduce a least squares problem that uses a factorization of the matrix into a product UU^T and a certain quadratic representation of the noise with two vectors. (This is apparently the motivation of the term double over-parametrization.) They study the implicit bias that running gradient descent on this functional introduces. They show that under some assumptions gradient descent indeed converges to the minimizer of a certain functional that incorporates a nuclear norm term (promoting low rank) and an l_1-term (promoting sparse noise). They demonstrate the effectiveness of this approach via numerical experiments.

Strengths: The paper makes progress in the understanding of overparametrization and implicit bias of gradient descent. Moreover, it provides a new method for robust low rank matrix recovery. The paper incorporates some recent findings and techniques. The focus is clear and the flow is smooth. The experiments are illustrative. In the supplementary files the authors even provide the code for their experiments.

Weaknesses: The authors restrict to commuting measurements in their main result. While previous works have also made this restriction, it seems to rule out most cases of interest. It would be good if the authors could discuss at least 1-2 examples of commuting measurements, ideally appearing in practice. While the result itself appears to be novel and important, the proof seems to be quite similar to the reference [7] by Arora et al. It would be good if the authors could discuss commonalities and differences of their proof to [7].

Correctness: Yes. The methodologies for proof and experiments are reasonable.

Clarity: The content is clear and the paper is well-written.

Relation to Prior Work: The paper introduces a clever combination of two methods. The relation of previous contributions is discussed adequately.

Reproducibility: Yes

Additional Feedback: To be honest, I find the term "double overparametrization" a bit strange. I would still call it simply "overparametrization". Perhaps, the authors could think about this point and potentially adjust. I would suggest that the authors briefly discuss the following point which is sometimes overlooked when discussing implicit bias of gradient descent in the context of low rank matrix recovery. When additional restricting to positive semidefinite matrices it turns out that the original low rank matrix is often the UNIQUE solution to the linear equation y=A(X) that is positive semidefinite, see the paper "Implicit regularization and solution uniqueness in over-parameterized matrix sensing" by Geyer et al., arxiv:806.02046, for details. In such cases, it does not make much sense to speak about implicit bias and it is not surprising that gradient descent converges to this unique solution. (It is not completely clear to me, however, whether commutativity rules out such uniqueness.) In the context of the present paper, however, it seems that uniqueness of the solution does not hold, due to the additional terms representing the sparse noise. (However, if the noise terms are fixed, then due to positive definiteness, the solution may be unique in many cases of interest.) Another point: Section 2.2, last sentence: a reader might misinterpret this and the previous sentences in the sense that it is a weak point of (8) that one needs to choose the parameter lambda in the correct way. However, the right choice \lambda = 1/\sqrt{n} does not depend on unknown signal characteristics, so that this is fine. In the end, the gradient descent needs to choose the step size \alpha = 1/\lambda so it does not escape this problem, but again it is not a real problem because \alpha=1/\sqrt{n} is known to be the right choice. I would suggest to slightly adjust the text in order to avoid misunderstandings. ----------------- Comments after reading the author feedback and other reviews: I still think that this is a very good paper. However, my comment about the assumption of commuting measurements is not yet appropriately answered. I did not mean that they should specify a basically equivalent definition (joint diagonalization), but I was interested in a practical example. I am still not convinced that commuting measurements can be useful in practice. Also, I think that the commutativity assumption not only simplifies the analysis, but in fact "assumes away" most of the difficulties and the non-commuting case will actually be MUCH harder and possibly even much different.


Review 3

Summary and Contributions: The paper proposes an algorithmic regularization via the gradient descent (GD) (6) to solve the optimization problem (4) explicitly and the problem (8) implicitly in low-rank matrix recovery. The two problems are connected to each other by setting \alpha=1/\lambda, which is justified in Theorem 1. Numerical simulations show that the proposed algorithm (6) outperforms conventional algorithms in robust PCA and robust recovery of natural images.

Strengths: The main contributions are twofold: One is a doubly over-parameterized formulation (4). The other contribution is Theorem 1 claiming a connection between the two optimization problems (4) and (8). The proof of Theorem 1 needs to be improved, as pointed out in "correctness." Nonetheless, I believe that Theorem 1 is correct.

Weaknesses: The proposed algorithm requires a sufficiently small learning rate \tau in (6) in principle because Theorem 1 depends heavily on the continuum approximation in the limit \tau\to0. Nonetheless, Theorem 1 assumes that the GD (6) solves a global solution. These two assumptions might conflict with each other in practice.

Correctness: The current proof flow of Theorem 1 is confusing since the paper attempts to prove a global optimality by using a KKT "necessary" condition in Lemma 1. More precisely, we need a KKT sufficient condition: if a solution satisfies KKT conditions, then the solution is a global optimum. Nonetheless, I believe that this issue can be fixed by proving that the problem (8) is convex.

Clarity: Yes in the mainbody. No in the supplementary materials. The latter needs to be improved.

Relation to Prior Work: Yes.

Reproducibility: Yes

Additional Feedback: The authors addressed all comments I pointed out in Weaknesses. Thus, I keep my high score. ---- --Note the meaning of \circle in (4). Is it the Hadamard product? --Improve the presentation of the proof of Theorem 1. One option is to summarize the proof strategy of Theorem 1 at the beginning. Then, technical lemmas should be stated. Clarify where the lemmas are used. The proofs of the lemmas may be placed after the proof of Theorem 1. Finally, use the lemmas to prove Theorem 1.


Review 4

Summary and Contributions: This paper showed that how the implicit bias of gradient descent can be extended for robust low-rank matrix recovery, while avoiding overfitting. Authors proposed a double over-parameterization (DOP) for both the imposed model structures, namely the low-rank and sparse components. The proposed DOP formulation, algorithm, and application to robust recovery of natural images have been explained with a convergence analysis. Experimental results of (1) low-rank matrix robust recovery and (2) the robust recovery of images with varied salt-and-pepper noise (i.e., spatially sparse corruption) have been reported.

Strengths: The proposed double over-parameterization (DOP) method for robust low-rank matrix recovery aims to overcome the challenge of blindly estimation of the underlying rank and sparsity level of the oracle data, if such prior knowledge is unknown. While existing works proposed and demonstrated the use of implicit bias of gradient descent on over-parameterized model for blind (without knowledge of rankness) low-rank recovery (without sparse error), the reliable extension to robust recovery of low-rank matrix recovery is somewhat new. Authors presented the problem formulation, algorithm with the control of the implicit regularization, as well as convergence analysis. The results are shown to be practically useful for salt-and-pepper image blind denoising problems, and demonstrated relative strength over other unsupervised learning methods, such as DIP.

Weaknesses: Reviewer is not exactly in this field, thus may not be in a good position to comment all aspects. Novelty: Since existing works have already proposed the over-parameterization in low-rank recovery problem [5,6], it is natural to impose the similar over-parameterization to the sparse modeling (which is also proposed in [2,3]) when generalizing to robustness low-rank matrix recovery. The combination of [2,3] with [5,6] seems pretty straightforward, thus the novelty is that significant. Experiment: It is unclear why only the DIP based methods are chosen to be the competing methods in the experiment part. Even if the experiments are under the blind (i.e., unknown noise level) setting, there are other deep learning approaches for solving such problems, as well as running noise estimation methods explicitly. The experiment results seem less convincing by choosing only few selected baselines, such as DIP, whose results highly depends on the early stop which is hard to judge whether the algorithm has been tuned to be optimal.

Correctness: The description and explanation are overall correct.

Clarity: Yes, the presentation is clear, while the experiment part can be strengthen with more justification on the result evaluation and competing methods.

Relation to Prior Work: Yes, the related low-rank recovery and over-parameterization methods are discussed, such as [2,3], and [5,6].

Reproducibility: Yes

Additional Feedback: I am satisfied with the authors' rebuttal, and "up-scaled" my score.

[Author Response · NeurIPS 2020]

We thank the reviewers for their detailed and thoughtful comments. We are encouraged that all reviewers (R1 - R4) find
our double over-parameterization approach for robust recovery problems to be novel and appreciate our theoretical
analysis of the gradient flow dynamics associated with the proposed formulation. The reviewers think our work help
advance the understanding of over-parameterization and implicit bias of gradient descent (R2) which may bear insight
into other related works (R1). Moreover, the reviewers find the experiments to be illustrative (R2) and have demonstrated
the relative strength of our approach over other unsupervised learning methods (R4).
All minor comments and corrections will be addressed in the final version. Implementation details can be found at line
249 and our submitted code. In the following, we address each reviewer's comments in detail one by one.

**Response to Reviewer 1.**

• *Q1: Natural images may not have low-rank structures.* A1: We did not model natural images by low-rank structures.
Rather, we modeled natural images by untrained deep networks following the work of DIP [34] (see line 46).
• *Q2: Denoising performance for other types of noise.* A2: This paper aims to provide a new framework for dealing
with overfitting and parameter tuning for robust learning in over-parameterized models, and our exposition adopts
sparse noise modeling *only* as a proof of concept. This opens new ways to handle other types of noise by redesigning
the over-parameterization term for noise accordingly, which is certainly a topic of interest for future work.
• *Q3: Needs practical guidance on the learning rate selection.* A3: One strength of our method is precisely that it
does *not* require a case-by-case selection of learning rate (see lines 11, 95, 218, 264).
• *Q4: Needs a brief discussion of the impact of sampling rate.* A4: If by "sampling rate" the reviewer meant the
sparsity level of the corruption term $s_\star$, its effect is demonstrated in our experiments (see line 241, 264). Or, if the
reviewer meant the sampling rate for the robust matrix recovery problem, we proved that it is the same as that of the
convex optimization approach which is information-theoretically optimal (see line 168).
• *Q5: More numerical comparisons with SOTA should be included.* A5: The main purpose of the paper is to address
the issue of overfitting. Nonetheless, our experiment already demonstrates superior performance when compared
with DIP - the SOTA unsupervised method. We will add more comparisons in the full version.
• *Q6: Computational time comparison and convergence behavior discussions are missing.* A6: The running time of
our method is comparable to DIP. The convergence behavior is discussed in line 258 and illustrated in Fig. 1 & 5.

**Response to Reviewer 2.**

• *Q1: Questions on commuting measurements.* A1: We adopt the commutative assumption to simplify the analysis.
One example of commuting measurements is when $\{\mathbf{A}_i\}_{i=1}^n$ are symmetric and jointly diagonalizable. Nevertheless,
there are both empirical [5,7] and theoretical [6] evidence showing that this assumption is not necessary. We leave
the study of recovery under more generic assumptions as interesting future work.
• *Q2: Commonalities and differences of the proof to [7].* A2: The work of [7] only handles low-rank matrix recovery
while our work handles both sparse and low-rank recovery. Although our proof follows a similar procedure as that in
[7], our contribution is on handling additional sparse terms and characterizing the discrepant learning rates in the
verification of the dual certificate. We will clarify this in the final version.
• *Q3: Uniqueness of solution.* A3: Thanks for the reference. We will cite it and add a discussion in the final version.

**Response to Reviewer 3.**

• *Q1: Conflicting assumptions on the learning rate $\tau$ between theory and practice.* A1: To simplify the analysis, we
worked in a asymptotic setting where $\tau \to 0$. In experiments (see Sec. 4.1), we showed that our method converges
non-asymptotically with a reasonably small learning rate (e.g., $\tau = 10^{-4}$). While we agree that there is a gap between
theory and practice, we believe that it can be addressed (e.g., by adapting the proof in [6]) and leave it to future work.
• *Q2: Correctness of proof to Theorem 1.* A2: We appreciate the reviewer's efforts in reading into the proofs and
pointing out a typo. Indeed, we intended to use the KKT as a *sufficient* condition (which holds by Slater's condition).

**Response to Reviewer 4.**

• *Q1: Novelty and significance.* A1: Our method is *not* a trivial or naive combination of low-rank and sparse
parameterizations. As explained in Sec. 2.3, it is crucial to design a proper learning algorithm, by means of implicit
bias of discrepant learning rates, to obtain correct recovery. With such a learning framework not only did we provide
theoretical justification but also demonstrated its good performance. We believe that this framework could have broad
implications beyond the robust matrix and image recovery problems: for many other problems in (deep) learning
(such as with label noise), it could help us to design scalable and principled ways to deal with epochwise overfitting.
• *Q2: Experimental comparisons.* A2: 1) Reason for using DIP as a baseline. Our method is an unsupervised approach
that performs single image denoising, and DIP is the best performing method in this category. We believe by "deep
learning approaches" the reviewer refers to supervised methods such as [32], which require extra training data and
cannot serve as a fair baseline. We will clarify this in the final version. 2) Fairness of comparison with DIP. Our
comparison is more than being fair: we granted DIP the privilege of using the ground truth clean image to determine
the best early termination, while only took the results for our method at its convergence that requires no access to the
ground truth. We have provided the code for the interested readers to verify our results.

[Meta-Review · NeurIPS 2020]

The paper has been discussed after the rebuttal that the reviewers found useful and actionable (e.g., clarification about the novelty of the proof and about the experiments). The paper is recommended for acceptance. All reviewers have acknowledged that the paper makes a step towards better understanding over-parameterization and the implicit bias of gradient descent. As promised in the rebuttal, it is important to include in the final version of the paper the mentioned clarifications and discussions.